# RNNs are not Transformers (Yet): The Key Bottleneck on In-Context Retrieval

**Kaiyue Wen**[1*]  **Xingyu Dang**[2*]  **Kaifeng Lyu**[3‡]
[1] Stanford University [2] Tsinghua University
[3] University of California, Berkeley
kaiyuew@stanford.edu   dangxy20@mails.tsinghua.edu.cn
kaifenglyu@berkeley.edu

## Abstract

This paper investigates the gap in representation powers of Transformers and Recurrent Neural Networks (RNNs), which are more memory efficient than Transformers. We aim to understand whether RNNs can match the performance of Transformers, particularly when enhanced with Chain-of-Thought (CoT) prompting. Our theoretical analysis reveals that CoT improves RNNs but is insufficient to close the gap with Transformers. A key bottleneck lies in the inability of RNNs to perfectly retrieve information from the context, even with CoT: for several tasks that explicitly or implicitly require this capability, such as associative recall and determining if a graph is a tree, we prove that RNNs are not expressive enough to solve the tasks while Transformers can solve them with ease. Conversely, we prove that adopting techniques to enhance the in-context retrieval capability of RNNs, including Retrieval-Augmented Generation (RAG) and adding a single Transformer layer, can elevate RNNs to be capable of solving all polynomial-time solvable problems with CoT, hence closing the representation gap with Transformers. We validate our theory on synthetic and natural language experiments.

## 1 Introduction

Transformers (Vaswani et al., 2017) have become the dominant choice of the backbone for Large Language Models (LLMs). The core component of Transformers is self-attention modules, which allow the model to route information densely across the entire sequence. However, this design leads to high inference costs for long sequences, including a memory cost linear in the sequence length to maintain intermediate attention keys and values for each token, and a time cost quadratic in the sequence length to compute the attention score for each pair of tokens.

Recently, Recurrent Neural Networks (RNNs) have become an increasingly popular choice in sequence modeling tasks due to their ability to maintain a memory size constant in sequence length during inference, thus being more memory efficient than Transformers. Katharopoulos et al. (2020) showed that Linear Transformers (Transformers with linear attention) can be expressed as RNNs. Gu et al. (2022) took a different path to design RNNs by structuring latent states as State Space Models (SSMs) from control theory. These ideas have led to a series of development of modern RNNs, including RWKV (Peng et al., 2023), RetNet (Sun et al., 2023), and Mamba (Gu & Dao, 2023). Most notably, Mamba can achieve competitive performance with Transformers on several sequence modeling tasks with linear time and constant memory in sequence length.

*Can RNNs replace Transformers yet?* The rise of these modern RNNs has led to an interest in understanding their limitations. A recent work by Arora et al. (2023) showed that a broad family of RNNs, input-independent gating SSMs, are empirically inferior to Transformers in a task that has a long history in artificial intelligence, *Associative Recall* (AR) (Willshaw et al., 1969; Hopfield, 1982; Hinton & Anderson, 2014): Given a series of key-value pairs as a string, the model is required to recall the value given a key. On the theory side, Sanford et al. (2023) and Jelassi et al. (2024) demonstrated that constant-memory RNNs do not have sufficient representation power to solve

---

*Equal contribution
†Corresponding author
‡The work was done while Kaiyue was at Tsinghua University and Kaifeng was at Princeton University.

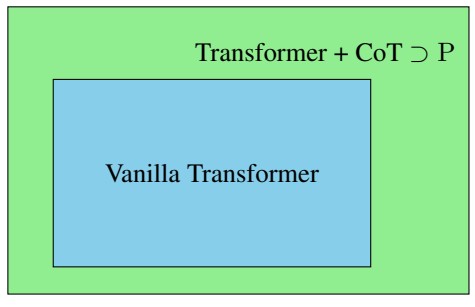 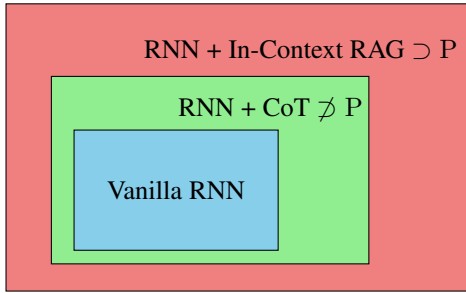

Transformer                                   RNN (This work)

Figure 1: **Hierarchy of Representation Power**. While RNN with chain-of-thought (CoT) with $O(\log n)$ bit memory provably has strictly stronger representation power than RNN without CoT under mild complexity assumptions (Theorem 4.1), it is still exponentially weaker than Transformer with CoT in representing solutions to algorithmic problems (Theorem 4.7). We proceed to show that the incapability of RNNs in In-Context Retrieval is the root cause of the gap and propose two forms of In-Context Retrieval Augmented Generation (In-Context RAG) to close the gap by illustrating their power to simulate any polynomial-time Turing machines (Theorems 5.2 and 5.4).

the tasks of averaging a given subset of input vectors ($q$-sparse averaging) and repeating the input sequence (copying), respectively, while there exist shallow Transformers that can solve these tasks.

However, the above results do not exclude the possibility that enhancing RNNs with additional prompting techniques or minor architectural changes could close the gap with Transformers. In fact, Transformers themselves are not perfect either and sometimes need additional techniques at inference time to perform well. For instance, Transformers may struggle with mathematical and algorithmic reasoning problems if they are forced to produce the correct answer immediately after processing the input sequence. But with *Chain-of-Thought* (CoT) prompting applied, a prompting technique that guides the model to generate a series of intermediate tokens before arriving at the final answer, their performance can be significantly improved. Feng et al. (2023); Li et al. (2024) explained the success of CoT from the perspective of representation power: Transformers alone do not have sufficient representation power to solve problems beyond a certain circuit complexity class ($\mathrm{TC}^0$), but with CoT, they can even simulate any polynomial-time Turing machine.

The effectiveness of CoT on Transformers naturally leads to the following question:

*Can similar enhancements, such as adopting CoT, improve RNNs to be on par with Transformers?*

**Our Contributions.** This paper examines potential ways to close the gap in the representation powers of RNNs and Transformers (with softmax attention) on algorithmic problems. Through a series of lower and upper bound results, we show that CoT improves the representation power of RNNs, but to close the gap with Transformers, CoT alone is not enough to overcome a key bottleneck of RNNs: their inability to retrieve information from the context, which we call *in-context retrieval* for short.

We further illustrate that addressing this in-context retrieval bottleneck is sufficient to close this gap: RNNs can solve all polynomial-time solvable problems if adopting techniques to enhance the in-context retrieval capability, including involving *Retrieval-Augmented Generation* (RAG) and using *hybrid architectures*, such as appending a single Transformer layer.

Our main contributions are listed as follows:

1. **CoT improves RNNs but cannot close the representation gap with Transformers.** (Section 4)

   - On the positive side, we prove that CoT makes RNNs strictly more expressive under mild assumptions from circuit complexity ($\mathrm{PSPACE} \not\subset \mathrm{P/poly}$).

   - On the negative side, we show that adopting CoT is not enough to close the representation gap between RNNs and Transformers: the memory efficiency of RNNs fundamentally limits their ability to perform in-context retrieval, even with CoT. This point is made concrete by proving that RNNs with CoT cannot solve a set of fundamental algorithmic problems that directly ask for in-context retrieval, including associative recall.

   - We further exemplify that in-context retrieval can be implicitly required in tasks that appear unrelated, by proving the inability of RNNs to solve the classic problem of determining whether a graph is a tree (IsTree).

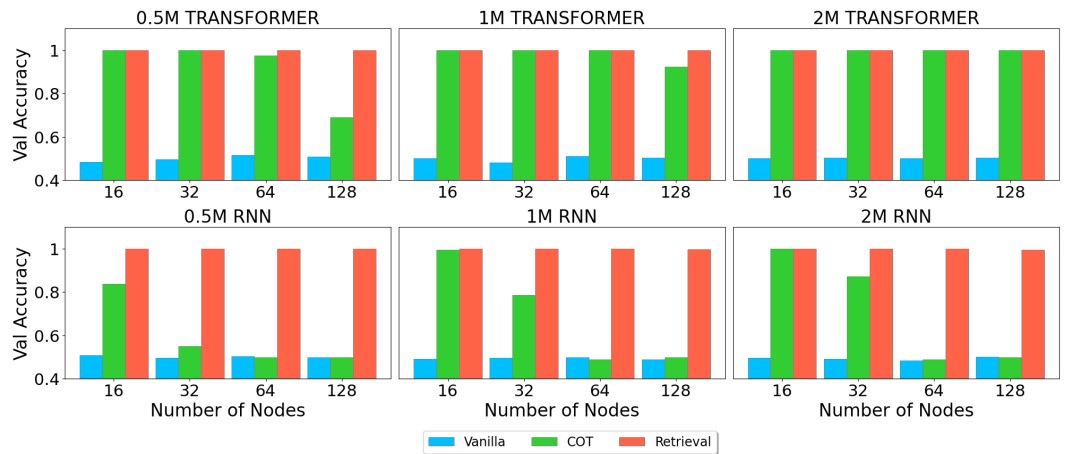

Figure 2: We train RNNs (Mamba) and Transformers (LLaMA 2 Touvron et al. (2023)) with a frozen word embedding and decoding head of three different model sizes (0.5M, 1M, 2M) on IsTree with three different sizes of graph (16, 32, 64) under three different setups. **Vanilla** means the model directly predicts the label. **COT** means the model will generate a chain-of-thought process based on DFS (see Algorithm 1) before prediction. **Retrieval** means the model will generate the chain of search queries and reasoning before prediction (see Algorithm 2). We observe that (1) Both Transformer and RNNs can't solve the IsTree question without a chain of thought; (2) RNNs' performance with chain-of-thought decays quickly when the number of nodes increase, which is consistent with our theory; (3) All models reach almost perfect accuracy when enhanced with retrieval.

Our negative results hold for a wide range of RNN architectures, including the aforementioned Mamba, RWKV, and even Linear Transformers. Technically, this is because RNNs are so memory efficient that they can trigger streaming lower bounds (Sanford et al., 2023), especially for problems that require in-context retrieval.

2. **Enhancing the in-context retrieval capability of RNNs can close the gap.** (Section 5)

   - We prove that allowing RNNs to invoke function calls to perform a primitive of in-context retrieval based on regular expression is sufficient to boost their representation power to solve all polynomial-time solvable problems with CoT, hence closing the representation gap.

   - As one layer of the Transformer is sufficient to perform many in-context retrieval operations, mixing some Transformer layers in RNNs should also narrow the representation gap. We prove that a minimal possible change in the RNN architecture can just work: adding one Transformer layer at the end of the RNN architecture is sufficient to close the representation gap.

   Our positive results showing that enhancing in-context retrieval can improve RNNs' representation power hold for vanilla Linear RNNs, and can be easily extended to more complex architectures. The intuition behind these results is that RNN can focus on the local reasoning steps and use the in-context retrieval module to adaptively fetch the relevant information from the context.

We validate our theoretical findings through synthetic and natural language experiments on IsTree and HotPot-QA, confirming that while CoT alone cannot close the performance gap between RNNs and Transformers, the proposed two solutions effectively narrow this gap. (Section 6)

**Implications.** We believe these results could provide valuable insights into architecture designs of LLMs: RNNs alone can suffer from many fundamental limitations in representation power, even with CoT; on the other hand, it is promising to explore strategies to enhance the in-context retrieval capability of RNNs with little overhead, such as using a hybrid architecture that mixes in one or more Transformer layers (Ren et al., 2024; Waleffe et al., 2024; Lieber et al., 2024b).

## 2 RELATED WORKS

**State Space Machines and Linear Transformers.** There has been a recent surge of interest in state space machines and (kernalized) linear transformers (Gu et al., 2022; Katharopoulos et al., 2020; Peng et al., 2023; Sun et al., 2023; Gu & Dao, 2023; Fu et al., 2023; Poli et al., 2023; Luo et al., 2021; Peng et al., 2021; Wang et al., 2020), which are a class of models that combine the parallelizability of the Transformer with the memory efficiency of the RNN. These models can process both a sequential and a recurrent form, and can use the former for fast parallelizable training

and the latter for memory-efficient inference. However, these models are still empirically inferior to the Transformer in terms of performance. Our work investigates the reasons behind this gap and proposes to close it by enhancing the in-context retrieval capability.

**Chain of Thought (CoT).** Chain of thought (Wei et al., 2023; Nye et al., 2021; Kojima et al., 2023; Wang & Zhou, 2024) is an augmentation to the Transformer, that allows it to solve more complex reasoning tasks by generating a reasoning process before outputting the answer. It has been shown that Transformers with CoT provably have more expressive power than the original Transformer without CoT (Feng et al., 2023; Li et al., 2024). However, the expressive power of RNNs with CoT has not yet been systematically studied. Theorem F.1 in Feng et al. (2023) shows that RNN cannot output a particular format of CoT for evaluating arithmetic expressions and solving linear equations while Transformers with the same amount of parameters can. Concurrent work (Yang et al., 2024) discovers that linear Transformers, a special class of RNNs, are not able to solve some dynamic programming problems with CoT, unless the number of parameters grows with the length of the input. One high-level message our work conveys is similar to theirs: RNNs have limited representation power to perform reasoning with CoT. However, we show that such limitation is not specific to the output format or architecture and apply tools from streaming complexity to prove lower bounds on a broader range of tasks and memory-efficient architectures.

**Streaming Algorithms.** Our lower bound leverages the technique in streaming algorithms. Streaming algorithms are algorithms that take constant (typically just 1) pass over the input and use sublinear space, hence including RNNs with fixed state space as a special case. Works in streaming algorithms date back to the 1980s (Munro & Paterson, 1980) and have been formalized and popularized in the 1990s (Alon et al., 1996) due to the need to process large data streams. The study of streaming algorithm is closely connected with the concept of **communication complexity**. The communication complexity of an algorithm is defined by the amount of communication cost required when the algorithm is distributed, and all streaming algorithms can be viewed as distributed algorithms with sublinear communication complexity, whose communication content is the internal state of the algorithm. The lower bound in our work is a direct application of this observation to the study of RNNs and we mainly consider the lower bounds for (1) indexing the input (Munro & Paterson, 1980) and (2) determining whether the input is a tree (Henzinger et al., 1998).

We will defer other related works to Appendix A.

## 3 PRELIMINARIES

We briefly introduce the definitions that are necessary for understanding our results and defer detailed definitions to Appendix B.

**Vocabulary and Embeddings.** Let $n$ be a parameter so that the input length can be bounded by $\Theta(n)$. We assume a finite vocabulary $V$ that includes natural numbers from 1 to $n$ and a constant number of special tokens. For our theoretical analysis, we fix word and position embeddings and only consider the representation power of the model when the non-embedding parameters can change. See Appendix B.3 for the formal definitions. Note that in the common practice, the vocabulary size does not increase with $n$ and numbers may be tokenized into a few tokens according to their decimal representations. Nevertheless, our lower bounds can be easily extended to this more practical case, since these results do not rely on the specific form of the vocabulary and embeddings. For upper bounds, our embedding scheme should be easily implemented from other embedding schemes with a few RNN or Transformer layers.

**Language Models.** We denote the set of all finite sequences and all non-empty finite sequences of tokens in $V$ as $V^*$ and $V^+$. A language model (LM) is a function $M : V^+ \to \mathbb{P}_V$ that maps a non-empty sequence to a probability distribution over the next token, where $\mathbb{P}_V$ is the probability simplex over $V$. We focus on LMs realized by deep neural networks: mapping an input sequence $\mathcal{S}$ to a sequence of embeddings $\text{Emb}(\mathcal{S})$, then applying a neural network (e.g., Transformer or RNN) to process the embeddings and output the probability distribution.

**Numerical Precision.** We consider models with fixed numerical precision. We use $p$ to denote the precision of the number of bits to represent real numbers and use $\mathbb{R}_p$ to denote the set of all real numbers that can be represented by $p$-bit floating point numbers. We defer the details to Appendix B.2. We assume $p = O(\log n)$, which is a common assumption when studying the finite precision neural networks (Feng et al., 2023; Merrill & Sabharwal, 2023).

**Transformers.** We study Transformers with a standard decoder architecture containing softmax attention and gated feedforward layers. See Appendix B.3 for definitions. We focus on constant-size Transformers, where the depth, width and number of parameters are fixed and do not depend on $n$.

**RNNs.** Recently there has been a lot of interest in the linear-time Transformer, which replaces the full-attention calculation with linear-time alternatives. These variants are mostly special forms of recurrent neural networks (RNNs) that are parallelizable. Here we define a general form of RNNs containing all the common variants to the best of our knowledge, including Mamba (Gu & Dao, 2023), RWKV (Peng et al., 2023), RetNet (Sun et al., 2023), StreamingLLM (Xiao et al., 2023), RMT (Bulatov et al., 2022), TOVA (Oren et al., 2024), xLSTM (Beck et al., 2024) etc. An RNN maintains a state $s_t \in \mathbb{R}_p^\Lambda$ storing $\Lambda$ $p$-bit floating point numbers. At each step, the RNN generates the next state $s_{t+1}$ based on the current state $s_t$ and token $\mathcal{S}_t$, and then it makes a prediction based on the updated state $s_{t+1}$. We characterize the complexity of an RNN with the following three measures: (1) the parameter size $P$ in bits, which is the total number of bits in parameters; (2) the memory size $M = \Lambda p$ in bits, which is the total number of bits in each state; (3) the circuit size $C$, which is the number of bit-wise operations needed to compute one step of the RNN. A particularly interesting class of RNNs is constant-size RNNs, where the dimension of state and number of parameters are fixed and do not depend on $n$, i.e., $\Lambda = O(1)$, $p = O(\log n)$, $P = O(\log n)$, $M = O(\log n)$, $C = O(\log n)$. We do not assume a specific structure of the RNN when we want to prove impossibility results for RNNs, i.e., what RNNs cannot represent. But when we want to showcase what RNNs can represent, we focus on Linear RNNs, one of the simplest form of RNNs, so that our results are more likely to be generalizable to more complex RNNs. See Appendix B.3 for the definitions of all the above models.

**Algorithmic Problems.** An algorithmic problem is a problem that may be solved by an algorithm. In this paper, we focus on algorithmic problems $f : V^+ \to V_{\mathrm{A}}$ that asks for computing $f(\mathcal{S}_{\mathrm{in}})$ given a sequence of tokens $\mathcal{S}_{\mathrm{in}}$ as the input, where $V_{\mathrm{A}}$ is the set of possible answers. We say that an LM $M$ can (directly) solve an algorithmic task $f$ if, given the sequence $\mathcal{S}_{\mathrm{in}}$, the probability distribution $M(\mathcal{S}_{\mathrm{in}})$ for the next token is peaked at the correct output token $f(\mathcal{S}_{\mathrm{in}})$, i.e., $\arg\max_{j \in V} M(\mathcal{S}_{\mathrm{in}})[j] = f(\mathcal{S}_{\mathrm{in}})$.

**Chain-of-Thought Reasoning.** *Chain-of-Thought* (CoT) reasoning allows the LM to produce intermediate steps before the final output. Following Feng et al. (2023); Li et al. (2024), our paper studies the effectiveness of CoT reasoning in improving the expressiveness of LMs, and we allow the intermediate steps to be an arbitrary sequence of tokens. We say that an LM $M$ can solve an algorithmic problem $f$ with CoT if the following process terminates with a sequence ended with $f(\mathcal{S}_{\mathrm{in}})$. First, let $\mathcal{S}_0 = \mathcal{S}_{\mathrm{in}}$. For all $i \geq 0$, decode the next token $s_i^{\mathrm{next}} = \arg\max_{j \in V} M(\mathcal{S}_i)[j]$ from $M$, and append it to the current sequence $\mathcal{S}_{i+1} = \mathcal{S}_i \oplus s_i^{\mathrm{next}}$. If $s_i^{\mathrm{next}} \in V_{\mathrm{A}}$, then the process terminates with $\mathcal{S}_{i+1}$ with $i$ steps of CoT; otherwise the process continues. It is evident that if an LM can solve an algorithm problem with 0 steps of CoT, then an LM $M$ can (directly) solve the problem. In this case, we also say that the LM can solve the problem without CoT.

## 4 How Well Does CoT Improve the Representation Power of RNNs?

In this section, we aim to understand how well CoT improves the representation power of RNNs and how it compares with Transformers.

### 4.1 CoT Strictly Improves RNNs

First, we show a positive result that RNNs with CoT can solve tasks that RNNs without CoT cannot solve. This relies on a mild complexity assumption $\mathrm{PSPACE} \not\subset \mathrm{P/poly}$, i.e., not all polynomial space complexity problems can be solved by a polynomial-size circuit family. This has been widely believed to be true since $\mathrm{PSPACE} \subset \mathrm{P/poly}$ would imply severe complexity consequences (Karp & Lipton, 1980; Babai et al., 1991).

**Theorem 4.1.** *Assuming* $\mathrm{PSPACE} \not\subset \mathrm{P/poly}$*, there exists an algorithmic problem such that (1) there exist constant-size Linear RNNs that can solve the problem with polynomial length CoT; and (2) any constant-size regular RNNs cannot solve the problem without CoT.*

See Appendix E.5 for the proof. The key insight is that the representation power of RNNs without CoT is limited to shallow circuits of size $\mathrm{poly}(\log n)$, but RNNs with CoT can simulate $O(\log n)$-space Turing machines perfectly with $\mathrm{poly}(n)$ steps. This result is consistent with previous works on the benefit of CoT for Transformers (Feng et al., 2023; Li et al., 2024), which also prove based on mild complexity assumptions that Transformers with CoT have the representation power to simulate

polynomial-size circuits to solve all the problems in P but Transformers without CoT cannot. Here we rigorously prove that a similar benefit of CoT also applies to RNNs, but a key difference is that CoT cannot boost the representation power of RNNs to simulate every polynomial-size circuit family.

## 4.2 COT CANNOT CLOSE THE REPRESENTATION GAP WITH TRANSFORMERS

Now we proceed to understand whether CoT can make RNNs as expressive as Transformers. The answer turns out to be negative: RNNs, even with CoT, struggle to solve very simple algorithmic problems that require the capability of retrieving information from the current context, which we call *In-Context Retrieval* for short. This limitation is directly related to the memory efficiency of RNNs. For a model with at most $o(n)$ bits in memory, we can involve techniques from streaming complexity to prove impossibility results for problems requiring In-Context Retrieval.

### 4.2.1 SIMPLE PROBLEMS ON IN-CONTEXT RETRIEVAL

Here we list several simple algorithmic problems that directly test the in-context retrieval capability of the model, which turn out to be a good test-bed for understanding the limitations of RNNs compared to Transformers. We defer discussion on the history of these problems to Appendix C.

**Definition 4.2** (Index). Index is a problem that given a sequence of tokens with length $n$ and a query token $i \in [n]$, requires the model to output the type of the $i$-th token.

**Definition 4.3** (Associative Recall). Associative Recall (AR) is a problem that given a sequence of tokens with length $n$ consisting of tokens in $[n]$ and a query token $q \in [n]$, requires the model to output the next token of $q$ in the sequence.

**Definition 4.4** ($c$-gram Retrieval). An $c$-gram is a contiguous subsequence of $c$ tokens in a sequence of tokens. $c$-gram retrieval is a problem that given a sequence of tokens with length $n$ and a query $(c-1)$-gram that is the prefix of a $c$-gram in the sequence, requires the model to output the last token of that $c$-gram.

**Definition 4.5** (Counting). Counting is a problem that given a sequence of tokens with length $n$, a query token $q \in [n]$, and a query number $t \in \mathbb{N}$, requires the model to output 0 or 1 to indicate whether the number of occurrences of $q$ in the sequence is greater than $t$.

The following theorems show that constant-size RNNs cannot solve any of the four tasks when the context length is long, while constant-size Transformers can solve them perfectly.

**Theorem 4.6.** *For task $T \in \{Index, AR, c\text{-gram retrieval}, Counting\}$, there exist constant-size Transformers that can solve $T$. On the other hand, any RNN with $o(n)$-bit memory cannot solve $T$ of size $n$ with any length of CoT for large enough $n$.*

See Appendix E.4 for the proof. The key idea of the lower bound is to view RNNs in the framework of communication complexity and use information-theoretic arguments to prove a lower bound. The computation of RNNs can be distributed to two parties: party A simulates the RNN on the first part of the input and sends the state to party B, then party B receives the state and simulates the RNN on the second part of the input (the query string) until it finishes producing the output. Therefore, if the RNN can solve the problem with $o(n)$-bit memory, then the input can be compressed to $o(n)$ bits while maintaining all the necessary information to answer all possible queries, which contradicts certain information-theoretic lower bounds.

We note that Theorem 4.6 does **not** imply that RNNs are incapable of in-context retrieval at all. Instead, it states that the maximal context length that RNNs can effectively retrieve from is linear in its state size. Although for a short context window, RNNs can be trained to perform in-context retrieval (see e.g. Arora et al. (2023); Gu & Dao (2023)), this limitation in retrieval capabilities has been empirically observed: for example in Waleffe et al. (2024), both pretrained Mamba and Mamba-2 7B models are shown to have significantly worse Phonebook-retrieval capabilities on 1K context length than Transformers with the same size and trained on the same data.

### 4.2.2 LIMITATIONS OF RNNS BEYOND SIMPLE IN-CONTEXT RETRIEVAL PROBLEMS

A natural question would be if an algorithmic problem does not directly test the in-context retrieval capability, can we hope that RNNs would have the representation power to solve it? Do RNNs and Transformers have the same representation power in this case? Still, the answer is negative. We show

that the limited memory size in RNNs can still be a bottleneck in solving algorithmic problems. Even if the retrieval capability is not explicitly tested in an algorithmic problem, it may still be required implicitly for reasoning about the answer.

We demonstrate this gap on a simple algorithmic problem that is seemingly irrelevant to retrieval: the IsTree problem. Given an undirected graph $G$ of $n$ nodes, determine whether $G$ is a tree, i.e., whether every pair of nodes is connected by exactly one simple path. We convert the graph $G$ into a token sequence, where we input the graph edge by edge. Figure 6 shows an example of the IsTree problem. A classical solution to IsTree is running Depth First Search (DFS), which takes $O(n)$ time. See Appendix B for more details.

Our result below shows that constant-size RNNs cannot solve IsTree, even with CoT. But constant-size Transformers can perform CoT reasoning by simulating DFS and perfectly solve the problem.

**Theorem 4.7.** *There exist constant-size Transformers that can solve IsTree with CoT of length $O(n)$. On the other hand, any RNN with $o(n)$-bit memory cannot solve IsTree with any length of CoT.*

See Appendix E.6 for the proof. The key idea of the lower bound is to again utilize information-theoretic lower bounds. This idea lies in the core of streaming complexity literature and investigation on the IsTree problem, dating back to Henzinger et al. (1998). We hereby restate the proof for completeness.

Given any binary sequence $x$ of length $n - 2$ and an index $k \in [n - 3]$, we construct a graph as follows: the graph has $n$ nodes, and vertex $a$ is connected to vertex $x_a + n - 1$ for any $a \in [n - 2]$. Additionally, vertex $k$ is connected to vertex $k + 1$. The graph is a tree if and only if $x_k \neq x_{k+1}$. Assuming there is an RNN with $o(n)$-bit memory that can solve IsTree, consider two parties, A and B, each holding the sequence $x$ and the index $k$. They can construct two parts of the graph using their information: A simulates the RNN on the first part and sends the state to B, who then simulates the RNN (potentially with CoT) on the second part to determine the output of the IsTree problem, equivalent to checking whether $x_k \neq x_{k+1}$. However, since $k$ is never sent to A, B can determine whether $x_k \neq x_{k+1}$ for any $k \in [n - 3]$, contradicting certain information-theoretic lower bounds.

### 4.2.3 TRANSFORMERS (WITH COT) ARE STRICTLY MORE EXPRESSIVE THAN RNNS

The above theorems show that RNNs have a significant representation gap with Transformers, even with CoT. Finally, we show that this gap is one-sided: there only exist tasks where Transformers can solve them and RNNs cannot, but not the other way around. The proof (Appendix E.7) is inspired by a recent work on CoT for Transformer (Li et al., 2024).

**Theorem 4.8.** *Given input length $n$, let $R$ is an RNN with word embedding $W^{(E)} \in \mathbb{R}_p^{(n+n_S) \times d}$, where $p = \Theta(\log n)$ is the precision, the constant $n_S$ is the number of special symbols in the vocabulary, the constant $d$ is the embedding dimension. If each recurrent iteration can be computed by a circuit of size $C(n) \leq 2^{p/2}$, and if the RNN produces the final answer after running at most $n^A$ steps of CoT for some constant $A > 0$, then there exist Transformers with $O(\log n)$-bit precision, $O(C(n))$ parameters and word embedding $\begin{bmatrix} W^{(E)} & \mathbf{0}^{(n+n_S) \times d} \end{bmatrix}$ that can produce the same final answer after running $(C(n) + 1)n^A$ steps of CoT.*

## 5 ENHANCING THE IN-CONTEXT RETRIEVAL CAPABILITY CLOSES THE GAP

In Section 4.2, we show that RNNs are deficient at In-Context Retrieval, hence leading to a significant representation gap with Transformers. In this section, we aim to understand: if we enhance the In-Context Retrieval capability of RNNs, do RNNs remain to have any representation gap with Transformers? We answer this question by examining two representative approaches to enhance the In-Context Retrieval capability, one explicit and one implicit, and show that both ways can close the representation gap between RNNs and Transformers in solving algorithmic problems.

### 5.1 EXPLICIT RETRIEVAL THROUGH REGULAR EXPRESSION

First, we explore the power of RNNs with *Retrieval Augmented Generation* (RAG), which gives an LM the capability to retrieve relevant information to assist generation. In our context, we are specifically interested in allowing LMs to call functions to retrieve information from their context, which we call *In-Context Retrieval Augmented Generation* (In-Context RAG).

Perhaps the simplest form of In-Context RAG is to allow the model to invoke function calls to associative recall, but we show in Appendix E.8 that this is not enough to close the representation

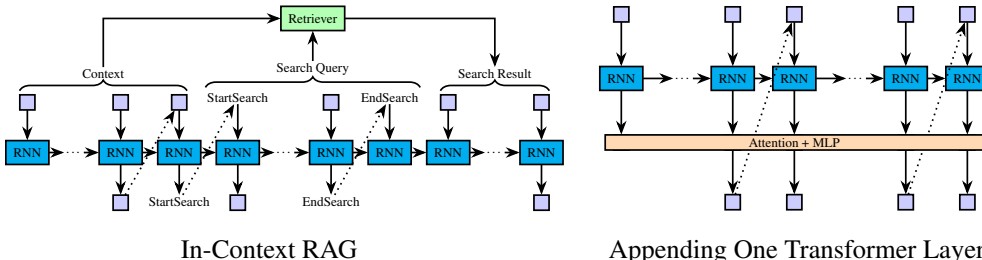

Figure 3: Illustration of the two approaches to enhance the In-Context Retrieval capability of RNNs: In-Context RAG (**left**) and appending one Transformer layer (**right**).

gap between RNNs and Transformers. In this light, we need to consider a more general form of In-Context Retrieval capability. We then turn to implement In-Context RAG by enabling an LM to perform regular expression matching, since the regular expression is a flexible primitive that can be used to describe a wide range of retrieval tasks and can be implemented efficiently on modern hardware. To be precise, when the context is string<StartSearch>pattern<EndSearch>, we evaluate the Python code `re.search(pattern, string).group(1)` to get the result of the regular expression matching. Then we append the result to the end of the context. See Definition B.10.

The following theorem shows that In-Context RAG with regular expressions is powerful enough for RNNs to solve the In-Context Retrieval problems in Section 4.2.1 with CoT.

**Theorem 5.1.** *For task $T \in \{$Index, AR, c-gram retrieval, Counting, IsTree$\}$, there exist constant-size Linear RNNs that can solve $T$ with In-Context RAG. For $T$ other than IsTree, $O(1)$ steps of CoT is required and for IsTree, $O(n \log n)$ steps of CoT is required.*

More generally, the following theorem further shows that In-Context RAG empowers constant-size RNNs to simulate polynomial-time Turing machines.

**Theorem 5.2.** *Given $A, B > 0$, for all polynomial-time Turing machines $T \in \text{TIME}(n^A)$ with $B$ states and vocabulary size $B$, there exist Linear RNNs with $B$ special symbols, $O(A \log n)$-bit precision and memory, and $O(AB^2)$ parameters that can output the result of $T$ by running $O(n^A)$ steps of CoT with In-Context RAG.*

By introducing In-Context RAG, RNNs are able to use regular expressions to retrieve information from a distance and put relevant information together. Then RNNs can use perform more complex operations locally and output tokens to be retrieved later.

As a final note, our focus here is to understand the representation power of RNNs given an appropriate RAG, but not to propose a method that immediately leads to practical applications. While the above results show that In-Context RAG can close the representation gap between RNNs and Transformers in solving algorithmic problems, a limitation here is that In-Context RAG is not an immediate practical solution, as there is no existing training data for this In-Context RAG.

## 5.2   IMPLICIT RETRIEVAL BY APPENDING JUST ONE TRANSFORMER LAYER

Since Bahdanau et al. (2016), attention mechanisms have been understood as a form of compensation for the fixed memory size of RNNs, allowing the model to attend to the entire context and retrieve information. More recently, several hybrid Transformer-RNN architectures, such as Jamba (Lieber et al., 2024a), SPADE (Zuo et al., 2022), and Block-state Transformers (Fathi et al., 2023), have shown significant performance improvements over RNNs in practice.

Now we demonstrate that adding attention layers as a form of implicitly enhancing the retrieval capability can close the representation gap. We study a simple hybrid architecture that combines RNN and Transformer by appending just one Transformer layer to the RNN output (Appendix B). The following theorem shows that it can solve the In-Context Retrieval problems in Section 4.2.

**Theorem 5.3.** *For task $T \in \{$Index, AR, c-gram retrieval, Counting, IsTree$\}$, there exist constant-size hybrid Linear RNNs that can solve $T$. For $T$ other than IsTree, no CoT is required and for IsTree, $O(n \log n)$ steps of CoT is required.*

Further, we show that this hybrid architecture is powerful enough to even simulate all polynomial-time Turing machines with CoT. The proof idea is similar to the case of using In-Context RAG but now we use attention to retrieve information instead.

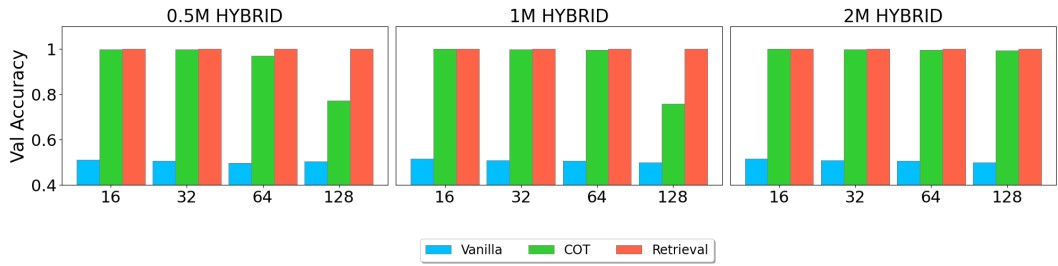

Figure 4: We train Mamba with one additional Transformer layer with a frozen word embedding and decoding head of three different model sizes (0.5M, 1M, 2M) on IsTree with three different sizes of graph (16, 32, 64) under three different setups. Vanilla means the model directly predicts the label. COT means the model will generate a chain-of-thought process based on DFS (see Algorithm 1) before prediction. Retrieval means the model will generate the chain of search queries and reasoning before prediction (see Algorithm 2).

**Theorem 5.4.** *Given $A, B > 0$, for all polynomial-time Turing machines $T \in \text{TIME}(n^A)$ with $B$ states and vocabulary size $B$, there exist a constant-size hybrid Linear RNNs with $B$ special symbols, $O(A \log n)$-bit precision and memory, and $O(AB^2)$ parameters that can output the result of $T$ by running $O(n^A)$ steps of CoT.*

## 6 EMPIRICAL VALIDATION

We validate our theoretical findings through synthetic and natural language experiments: CoT alone cannot close the performance gap between RNNs and Transformers, but enhancing the in-context retrieval capability can narrow the gap.

### 6.1 VALIDATION ON SYNTHETIC TASK: ISTREE

First, we validate our theoretical findings on the IsTree task introduced in Section 4.2.2. We generate a synthetic dataset for this task, where each data point is a tokenized graph concatenated with an answer of YES or NO, potentially with a CoT inserted in between. We then train RNNs and Transformers as language models on this dataset autoregressively.

**Experiment Details.** To generate the graph, we follow the procedure described in the proof of Theorem 4.7 (see Figure 6). The CoT data is generated using Algorithm 1 and the retrieval data is generated using Algorithm 2. For the CoT model, we decode the reasoning path during inference time until we reach the first YES or NO up to a max token limit greater than the length of all ground truth CoT. For the data points that the model fails to give a prediction, we assume the model gets it correct with 0.5 probability. For the retrieval task, we omit the explicit format of the regular expression and only ask the model to generate the vertices and special tokens in the regular expression to shorten the length of the input sequence. The reported accuracy is calculated over a validation set of 5000 samples using the last iteration of the model. We defer the experiment details to Appendix D. The results are shown in Figures 2 and 4.

**Results with CoT.** Without CoT, both the Transformers and RNNs model cannot learn to solve the IsTree problem. CoT improves the performance of both Transformers and RNNs but the RNNs' performance degrades sharply as the graph size increases and the Transformers consistently outperforms RNNs in this case. This is consistent with our theory that CoT can improve the expressiveness of the RNN models but the expressiveness is still not enough to solve IsTree (see Theorems 4.1 and 4.7).

**Results with In-Context RAG.** In-Context RAG allows all the models to reach near-perfect accuracy. This is consistent with our theory that retrieval augmentation via regular expression can improve the expressiveness of the RNN models to solve algorithmic tasks (see Theorems 5.2 and E.33). The hybrid model (a Mamba model with one additional Transformer layer on the top) shows performance on par with the Transformer, which is consistent with our theory (see Theorems 5.4 and E.36).

### 6.2 VALIDATION ON HOTPOT-QA

We further conduct experiments on open-source LLMs based on Transformer, Mambda, and a hybrid architecture to show that for tasks that require stronger in-context retrieval capability, RNNs suffer more performance degradation compared to Transformers.

**Experiment Details.** We use Phi-1.5 1.3B (Li et al., 2023) as our Transformer model. We further use two Mamba and Transformer-Mamba hybrid models distilled from Phi-1.5 with approximately the

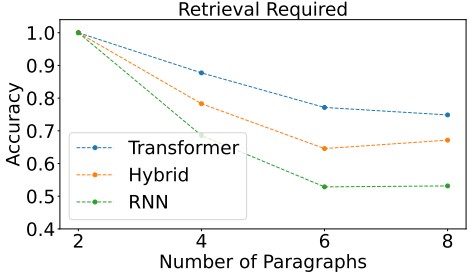 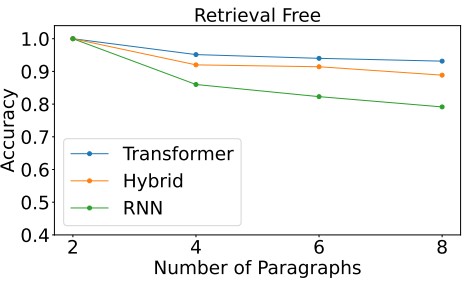

Figure 5: **Evaluation of Multi-Document Reasoning on Hotpot-QA.** $n$ paragraphs are given to the model before the question and 2 among which contains required information to answer the question. (Left). When provided paragraphs are given in a random order, the task requires stronger in-context retrieval capabilities when $n$ increases. RNNs' performance drop more sharply, as predicted by theory. (Right). When the required paragraphs are always at the end of the provided paragraphs, the retrieval difficulty significantly decreases and all the models have stable performance with respect to $n$.

same parameters' sizes (Bick et al., 2024) as our RNN and hybrid models. To test our theory, we use the Hotpot-QA (Yang et al., 2018) dataset. In the validation set of Hotpot-QA, each question is accompanied by a set of different related paragraphs from Wikipedia. 2 of these paragraphs contain useful information to answer this question. The model needs to retrieve the correct paragraphs and reason based on these paragraphs to get the answer to the question.

**HotpotQA with Controlled Retrieval Difficulty.** We consider the following version of Hotpot-QA with enhanced retrieval difficulty: We choose $n$ paragraphs containing the correct paragraphs and order them randomly before the question. The model needs to answer the question based on the given paragraphs after CoT reasoning. This design allows us to use the hyperparameter $n$ to control the difficulty of in-context retrieval in this task, with larger $n$ corresponding to a higher difficulty. We test our models under a 4-shot setting with Chain-of-Thought. The model we tested has varying performance even if $n = 2$. This is mostly due to their different capabilities to follow instructions and in-context examples. To mitigate this effect and highlight the impact of retrieval, we only test on a subset of $350$ samples of the validation set where all the models can answer correctly given the correct paragraphs. This ensures perfect accuracy when $n = 2$ in both settings. We vary $n$ in $\{2, 4, 6, 8\}$ and the result is shown in Figure 5.

**RNNs' performance drops sharply with increased retrieval difficulty.** While all the model's performance drops with increased retrieval difficulty (Figure 5, left), the RNN model has the largest drops. The hybrid model with only $4$ attention layers performs significantly better than the RNN model. This validates our theory that RNN architectures' limited in-context retrieval capabilities will impact their performance in reasoning and hybrid architecture is a potential solution to this limitation.

**Deconfouding the effect of context length.** The context length also increases with the number of provided paragraphs and can be a potential confounder. We then experiment with a controlled group: after choosing the same set of $n$ paragraphs, we always order the correct paragraph at the end, which significantly simplifies the in-context retrieval process. In this case, all the models have stable performance when the number of paragraphs increases.

## 7 CONCLUSION AND DISCUSSION

This paper studies the representation gap between RNNs and Transformers on algorithmic problems. We prove that while CoT improves RNNs, it is insufficient to close the gap with Transformers due to the key bottleneck of RNNs' inability to perform in-context retrieval. We show that adopting In-Context RAG or appending a single Transformer layer to RNNs can enhance their in-context retrieval capabilities, thus closing the representation gap with Transformers.

Considering the inherent limitations of RNNs in in-context retrieval, it is crucial to explore strategies that enhance this capability. This includes developing more effective in-context RAG methods tailored for RNNs, beyond the theoretically simple but impractical approach using regular expressions discussed here. Further research should also focus on architectural modifications that strike a balance between representational power and memory efficiency. In this study, we demonstrate that adding just a single Transformer layer significantly enhances the representational capacity of RNNs. Future work could investigate the optimization and generalization of various hybrid architectures to achieve a more comprehensive understanding.

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

## CONTENTS

## A    EXTENDED RELATED WORKS

**Comparison Between Transformers and RNNs (without CoT).** A line of works focused on the comparison between RNNs and Transformers in terms of recognizing or generating formal languages (Bhattamishra et al., 2020; Hahn, 2020; Merrill et al., 2022). These works show that the lack of recurrent structure in Transformers makes them fail to recognize some formal languages that RNNs can recognize. However, Liu et al. (2023); Yao et al. (2023); Hao et al. (2022) show that such limitation can be mitigated when we consider bounded length of input or bounded grammar depth. Our work differs from these works in that we consider the expressive power of RNNs and Transformers with CoT and show that in this case, the gap between RNNs and Transformers is one-sided (Theorem 4.8). Another thread of recent works focuses on understanding why RNNs or SSMs are inferior to Transformers (Arora et al., 2023; Jelassi et al., 2024; Sanford et al., 2023; 2024). Our work differs from these prior works from two perspectives: (1) we identify the common reason behind a wide range of tasks where RNNs fall behind Transformers; (2) we show two potential solutions that can bridge the representation gap by addressing this common reason (the incapability to retrieve from context).

**Retrieval Augmented Generation.** Our work proposes to use retrieval augmentation to close the representation gap between RNNs and Transformers. This is consistent with the recent trend of retrieval augmented generation (Guu et al., 2020; Borgeaud et al., 2022; Rubin & Berant, 2023). Empirically, retrieval augmented generation has been shown to improve the performance of recurrent models in various tasks (Kuratov et al., 2024; Akyürek et al., 2024) and our work provides a theoretical foundation for this phenomenon. Our work also shows that an attention layer can be used to simulate the retrieval process, which is consistent with the finding that attention can improve the performance of RNNs (Vaswani et al., 2017; Arora et al., 2023; Park et al., 2024; Peng et al., 2023; Hao et al., 2019). It has also been shown empirically that attention can be used to simulate complex retrieval process (Jiang et al., 2022).

**Comparison Between Transformers and RNNs** Prior work (Arora et al., 2023) has shown that input-independent gating SSMs are inferior to Transformers in the task called *associative recall* (Willshaw et al., 1969; Hopfield, 1982; Hinton & Anderson, 2014). The task requires the model to recall a previously seen pattern given a partial input. They show that input-dependent gating SSMs have better performance in associative recall and also propose a hybrid architecture that combines input-independent state space machines with attention to achieve better performance. Our work differs from this work in the following ways: (1) Our work studies associative recall from a theoretical perspective and proves formal lower bounds on the memory size of RNNs necessary for solving associative recall and other retrieval tasks; (2) We also study hybrid architectures but we provide a proof that appending a single Transformer layer to RNNs can make them expressive enough; (3) Our theory applies to not only input-independent gating SSMs but also all RNNs with $o(n)$-bit memory.

Prior work (Jelassi et al., 2024) proves a representation gap between RNNs and Transformers in repeating a long sequence, which can be seen as a retrieval task. They show that RNNs have difficulty performing the task due to their limited memory. Our work further proves that RNNs are limited in solving many other retrieval tasks, even with CoT. Technically, a key ingredient in their proof is a counting argument on the output sequence to show a limited memory size is not enough to produce too many different output sequences, but our proof can handle retrieval tasks that only require outputting a single token.

Notably, Sanford et al. (2023) apply communication complexity to prove circuit size or memory size lower bounds for RNNs and Transformers on the task of sparse averaging. Sanford et al. (2024) extend this technique to another task called $\text{hop}_k$, a generalization of the associative recall task. Our technique is similar to theirs since our proof is also based on communication complexity. But we consider a broader range of tasks including seemingly irrelevant reasoning tasks such as IsTree, and further explore various ways to close the representation gap.

**Representation Theory of RNNs.** Another line of works (Li et al., 2021; 2022; Alberti et al., 2023) studies the universal approximation power of RNNs. They show that the upper bound of the approximation power of linear RNNs will be constrained by the dimension of the hidden states. Their works on the high level are consistent with our findings but are not directly comparable because we are considering finite precision compute models with the assistance of CoT or In-Context RAG.

# B ADDITIONAL DEFINITIONS

We will now define some definitions used in the proofs.

## B.1 REASONING TASKS ON GRAPHS.

When resaoning on graphs, without otherwise specified, we will use $n$ as the number of vertices and $m$ as the number of edges. Without loss of generality, we will assume the vertices are labeled by $[n]$.

We will focus on decision problems on graphs, which are defined as follows:

**Definition B.1** (Decision Problem on Graphs). A decision problem on graphs is a function $f : \mathcal{G} \to \{\text{YES}, \text{NO}\}$, where $\mathcal{G}$ is the set of all possible graphs.

We will use the following decision problem as our main example:

**Definition B.2** (IsTree). $\text{IsTree}(G) = \text{YES}$ if $G$ is a tree, and $\text{IsTree}(G) = \text{NO}$ otherwise.

One can view IsTree as a minimal example of reasoning task2s. One of the classical solutions to IsTree is running Depth First Search and this algorithm takes $O(n)$ time.

## B.2 MORE ON NUMERIC PRECISIONS.

We will use $\text{ROUND}(x, p)$ to denote the rounding function that rounds $x$ to the nearest number in $\mathbb{R}_p$. We will assume $p$ is an odd number without loss of generality.

$$\mathbb{R}_p = \left\{ (2b_p - 1) \left( \sum_{i=1}^{p-1} b_i 2^{(p-1)/2-i} \right) : \forall i \in [p], b_i \in \{0, 1\} \right\}. \tag{1}$$

For calculation over $\mathbb{R}_p$, we will assume the calculation is exact and the result is rounded to $\mathbb{R}_p$ at the end, that is, for operator $\oplus$, we will have

$$\text{ROUND}(x, p) \oplus_p \text{ROUND}(y, p)$$
$$= \text{ROUND}\left(\text{ROUND}(x, p) \oplus \text{ROUND}(y, p), p\right).$$

We will additionally define $\mathbb{Z}_p$ as the set of all integers that can be represented by $p$-bit floating point numbers. We will define $1/[m]$ as the set of unit fractional $\{\frac{1}{i}\}_{i \in [m]}$. Further, we will define $\text{ROUND}(1/[m], p)$ as the rounding of $1/[m]$ to $\mathbb{R}_p$. We will additionally define for any real number $x \in \{0, 1\}$, $\text{next}(x) = \frac{1}{m+1}$ where $m = \arg \min_{k \in \mathbb{Z}} |x - \frac{1}{k}|$.

## B.3 MODELS

**Tokenization.** To tokenize a string, we will tokenize all the words separated by the space character into a sequence of tokens. To tokenize a graph $G$, we will order its edges $\text{E} = \{(u_i, v_i) \mid u_i < v_i\}$ randomly and tokenize it into the following string:

$$\text{Tokenize}(G) = \{\texttt{}, u_1, \sim, v_1, \dots, u_m, \sim, v_m\}. \tag{2}$$

We hereby assume there are constant more special tokens that are not the same as any number token, which are listed below:

- $\texttt{}$: the first special token, indicating the start of a sentence.
- $\sim$: the second special token, indicating an edge.
- YES: the third special token, indicating the answer is yes.
- NO: the fourth special token, indicating the answer is no.
- $\langle\text{StartSearch}\rangle$: the fifth special token, indicating the start of a search query.
- $\langle\text{EndSearch}\rangle$: the sixth special token, indicating the end of a search query.

We will denote the total number of special tokens as $n_S$ and the total vocabulary size as $|V| = n + n_S$. We will further define the detokenization function Detokenize,

$$\text{Detokenize}(\mathcal{S}) = \text{``}\mathcal{S}_1 \, \mathcal{S}_2 \, \dots \, \mathcal{S}_l\text{''}.$$

Here each $\mathcal{S}_i$ is either a number or a special token, which we will treat as a word.

**Embedding Functions.** We will use $d$ to denote the dimension of the embedding and $w_i$ to denote the $i$-th coordinate vector in $\mathbb{R}^d$.

We will separate the embedding function into two parts: the word embedding and the position embedding. For the word embedding, we will use $iw_1 \in \mathbb{R}^d$ to represent the embedding of the vertice $i$ in the tokenization of the graph. For the $k$-th special token, we will use $w_{2+k}$ to represent its embedding. For example, the embedding of $\sim$ is $w_2$. We will denote the word embedding matrix as $W^{(E)} \in \mathbb{R}^{|V| \times d}$.

For the position embedding, we will use $lw_d$ to represent the position embedding of the $l$-th token in the tokenization of the graph, which is a hyperparameter. The final embedding of any token sequence is the sum of the word embedding and the position embedding. We will use Emb to denote the embedding function.

This embedding function will be fixed and shared across all models we consider in this paper and will not be learned during training, hence we will not consider it as part of the model parameters.

**Language Modeling.** We use $V^*$ and $V^+$ to denote the set of all finite sequences and all non-empty finite sequences of tokens in $V$, respectively. We study language models that can predict the next token given a prefix of tokens. For this, we define a language model (LM) as a function $M : V^+ \to \mathbb{P}_V$ that maps a non-empty sequence to a probability distribution over the next token, where $\mathbb{P}_V$ is the probability simplex over $V$. We specifically study the case where the language model is realized by deep neural networks: first map the input sequence $\mathcal{S}$ into a sequence of embeddings $\text{Emb}(\mathcal{S})$, and then apply a neural network, such as a Transformer or RNN, to process the embeddings and output the probability distribution. We would call a series of parameterized models with increasing input size a *family* of models.

**Transformer** We will first define the Transformer architecture used in the theoretical analysis in this paper.

**Definition B.3** (Transformer Block). Let $X \in \mathbb{R}^{d \times l}$ be the input matrix, where $l$ is the sequence length. The output of a Transformer block $f$ is defined as:

$$f(X) = X + \mathcal{A}(X) + g(X + \mathcal{A}(X)),$$
$$\mathcal{A}(X) = \sum_{h=1}^{H} W^{(V,h)} X \text{softmax} \left( \frac{\left(W^{(K,h)} X\right)^\top W^{(Q,h)} X}{\sqrt{d}} + C \right), \tag{3}$$

where $g$ is a column-wise ReGLU feed-forward network [1] with width $w$ and output dimension $d$, $\mathcal{A}$ is the scaled dot-product attention, $\text{softmax}$ is the column-wise softmax function, $W^{(K,h)}$, $W^{(Q,h)}$, $W^{(V,h)}$ are the learnable parameters and $H$ is the number of heads, and

$$C = \begin{bmatrix} 0 & 0 & \dots & 0 \\ -\infty & 0 & \dots & 0 \\ \vdots & \vdots & \vdots & \vdots \\ -\infty & -\infty & \dots & 0 \end{bmatrix} \in \mathbb{R}^{l \times l} \text{ is a mask to prevent the attention from attending to future}$$

tokens.

In the context of language modeling, given a sequence of tokens $\mathcal{S}$, a Transformer $T(\mathcal{S})$ is defined as:

**Definition B.4** (Transformer). Let $\mathcal{S} \in |V|^l$ be the tokenized input sequence, the output of a Transformer is defined as:

$$T(\mathcal{S}) = \text{softmax} \left( W^{(E)} \left( f_L \left( \dots f_1 \left( \text{Emb} \left( \mathcal{S} \right) \right) \right) \right) \right)_{:,l}. \tag{4}$$

where $\text{softmax}$ is the column-wise softmax function, $f_i$ is the $i$-th Transformer block. We will call the $i$-th Transformer block the $i$-th layer of the Transformer and denote its feed-forward layer and attention layer as $g_i$ and $\mathcal{A}_i$ respectively.

---

[1] ReGLU means $\sigma(x) = \text{ReLU}(W_1 x + b_1) \otimes (W_2 x + b_2)$, this is a surrogate for the commonly used SwiGLU activation and allows the model to perform multiplication of two coordinates.

**RNNs**  We formally define the RNN architecture here.

**Definition B.5** (RNN).  An RNN architecture is characterized by two functions: state transition function $\mathbf{t} : \Theta \to \left( \mathbb{R}_p^\Lambda \times \mathbb{R}_p^d \to \mathbb{R}_p^\Lambda \right)$ and output function $\mathbf{o} : \Theta \to \left( \mathbb{R}_p^\Lambda \to \mathbb{R}_p^d \right)$, where $\Lambda$ is the dimension of the state and $\Theta$ is the parameter space. Let $\mathcal{S} \in |V|^l$ be the input sequence, the output of a recurrent neural network with parameter $\theta \in \Theta$ is defined as:

$$R_\theta(\mathcal{S}) = \mathrm{softmax}\left( W^{(E)} \mathbf{o}_\theta(s_l) \right),$$

$$\forall k \in [l], s_k = \mathbf{t}_\theta(s_{k-1}, \mathrm{Emb}(\mathcal{S})_{:,k}),$$

where $s_0 \in \mathbb{R}_p^\Lambda$ is a vector determined by $\theta$ and $W^{(E)}$ is the word embedding matrix. We will omit the subscript $\theta$ when it is clear from the context.

We define the following RNN architecture for the proof of our upper bound.

**Definition B.6** (RNN block).  A Linear RNN block is defined as follows:

$$f(X) = X + \mathrm{LU}(X) + g(X + \mathrm{LU}(X)),$$

where $g$ is a column-wise ReGLU feed-forward network with width $w$ and output dimension $d$ and LU is a linear unit, defined as

$$h_0 = 0, h_{:,t} = A h_{:,t-1} + B X_{:,t}, \mathrm{LU}(X_{:,1:l}) = h_{:,1:l}.$$

**Definition B.7** (Linear RNN).  A Linear RNN is a recurrent neural network

$$R(\mathcal{S}) = \mathrm{softmax}\left( W^{(E)} \left( f_L \left( \dots f_1 \left( \mathrm{Emb}\left( \mathcal{S} \right) \right) \right) \right) \right)_{:,l}. \tag{5}$$

where $\mathrm{softmax}$ is the column-wise softmax function, $f_i$ is the $i$-th Linear RNN block. We will call the $i$-th Linear RNN block the $i$-th layer of the Linear RNN and denote its feed-forward layer and linear unit layer as $g_i$ and $\mathrm{LU}_i$ respectively.

**Hybrid RNNs.**  We formally define the hybrid RNN architecture here.

**Definition B.8** (Hybrid RNN).  A hybrid RNN is a model that consists of an RNN with transition and output function $\mathbf{t}, \mathbf{o}$ and one Transformer layer $f$, the output of the RNN is used as the input of the Transformer layer and the output of the Transformer layer is used to produce the next token. Concretely, given the input sequence $\mathcal{S}_{\mathrm{in}}$, the output of the hybrid architecture is:

$$\mathcal{H}(\mathcal{S}) = \mathrm{softmax}\left( W^{(E)} f \left( [\mathbf{o}(s_k)]_{k \in [l]} \right) \right)_{:,l},$$

$$\forall k \in [l], s_k = \mathbf{t}(s_{k-1}, \mathrm{Emb}(\mathcal{S})_{:,k}),$$

## B.4    Language Models for Reasoning.

**Chain of Thought.**  We will now define how we use language models to solve reasoning tasks utilizing the following technique called chain of thought.

**Definition B.9** (Chain of Thought).  Given a language model, $M$ with vocabulary $V$ and the tokenized input sequence $\mathcal{S}_{\mathrm{in}} \in |V|^{l_0}$, chain of thought (CoT) generates the following sequence of tokenized sequence:

$$\mathcal{S}_0 = \mathcal{S}_{\mathrm{in}}, \tag{6}$$

$$s_i^{\mathrm{next}} = \arg\max_{j \in V} M(\mathcal{S}_i)[j], \tag{7}$$

$$\mathcal{S}_{i+1} = \mathcal{S}_i \oplus s_i^{\mathrm{next}}, \forall i \geq 0. \tag{8}$$

The process terminates at $\mathcal{S}_i$ when $\arg\max_{j \in V} M(\mathcal{S}_i)[j]$ is YES or NO. The language model can solve the reasoning task within $T$ steps of CoT if the process terminates at $\mathcal{S}_i$ where $i \leq T$ and the final output is correct. We will call the special case where the language model solves the reasoning task within 0 steps of CoT as solving the reasoning task without CoT.

**Retrieval Augmentation.** We will show in this paper that retrieval augmentation is a necessary technique to solve reasoning tasks for recurrent neural networks. We will define retrieval augmentation as follows:

**Definition B.10** (Retrieval Augmented Generation). Given an LM $M$ with vocabulary $V$ (containing two additional special tokens, <StartSearch> and <EndSearch>) and the tokenized input sequence $\mathcal{S}_{\text{in}} \in |V|^{l_0}$, the LM $M$ with In-Context RAG generates following sequence of tokenized sequence:

$$\mathcal{S}_0 = \mathcal{S}_{\text{in}}, \qquad s_i^{\text{next}} = \arg\max_{j \in V} M(\mathcal{S}_i)[j],$$

$$\mathcal{S}_{i+1} = \begin{cases} \mathcal{S}_i \oplus s_i^{\text{next}}, & \text{if } s_i^{\text{next}} \neq \text{<EndSearch>} \\ \mathcal{S}_i \oplus s_i^{\text{next}} \oplus \text{RETRIEVE}\left(\mathcal{S}_i\right), & \text{otherwise.} \end{cases}$$

Here RETRIEVE looks for the last occurrence of <StartSearch> at position $l_s$ and <EndSearch> in $\mathcal{S}$ at position $l_e$ and treat $\text{Detokenize}(\mathcal{S}_{l_s:l_e})$ as a regular expression, where Detokenize maps the tokenized sequence back to the string, inserting a space between every pair of adjacent tokens. The algorithm then runs a regular expression matching on $\text{Detokenize}(\mathcal{S}_{1:l_s-1})$, finds the first matching substring, and returns the first capturing group according to the regular expression (i.e., content embraced by a pair bracket in the regular expression). While there are many grammar standards of regular expressions, we adhere to the standard specified in the `re` library of Python. That is, we evaluate the following Python code to get the result of the regular expression matching:

```
re.search(pattern, string).group(1)
```

where $\text{Detokenize}(\mathcal{S}_{l_s:l_e})$ is the pattern and $\text{Detokenize}(\mathcal{S}_{1:l_s-1})$ is the string.

We will note that assuming $|V| = O(n)$ and every search query and the result is of length $O(1)$, the regular expression evaluation can typically be evaluated in $O(n)$ time.

**IsTree** In the context of language modeling, we can write the graph $G$ as a sequence of tokens, and then the task of IsTree is to determine whether $G$ is a tree by predicting a YES/NO token with or without CoT. We use the following tokenization for the graph $G$:

$$\text{Tokenize}(G) = \{\text{}, u_1, \sim, v_1, u_2, \sim v_2, \ldots, u_m, \sim, v_m\}, \tag{9}$$

where  and $\sim$ are two special tokens representing the start of the sentence and an edge, and $u_i, v_i$ are numbers denoting the nodes of the graph.

## C  OMITTED DISCUSSION ON IN-CONTEXT RETRIEVAL EXAMPLES

Index and AR are perhaps the most basic problems in retrieval, where Index asks for retrieving a token from the input sequence viewed as a linear array of tokens, and AR asks for retrieving a token from the input sequence viewed as an associative array. These two problems have been studied extensively by different communities. Index is a classic problem in streaming and communication complexity (Munro & Paterson, 1980), known to be impossible to solve with $o(n)$ bits of memory for streaming algorithms. AR has been regarded as a fundamental problem that an artificial intelligence system should be able to solve (Willshaw et al., 1969; Hopfield, 1982; Hinton & Anderson, 2014; Graves et al., 2014; Ba et al., 2016). In the context of LLMs, AR has been observed to correlate with in-context learning performance (Elhage et al., 2021) and has also been used extensively as synthetic surrogate tasks for pretraining performance (Fu et al., 2023; Poli et al., 2023; Lutati et al., 2023). Besides Index and AR, $c$-gram retrieval is a natural extension of AR to the case where the query key can contain multiple tokens: instead of retrieving a token given a single-token key, $c$-gram retrieval asks for retrieving a token when the given key is a $(c-1)$-gram. This task has been studied empirically, but not theoretically in Jelassi et al. (2024). Counting is a problem that asks for the number of occurrences of a token, thereby testing the model's capability to retrieve some statistics of relevant information from the input sequence.

## D  OMITTED EXPERIMENT DETAILS

We train three different architectures: (1) LLaMA architecture (Touvron et al., 2023) representing Transformers, (2) Mamba architecture (Gu & Dao, 2023) representing RNNs, and (3) Mamba with

one additional layer of LLaMA block representing hybrid architectures. Following our theory, we freeze and weight-tie the prediction head and word embedding in all the models. For ease of training, we use a different embedding function mapping $i-$th token to $[\sin(\frac{i}{10000^{j/d}}), \cos(\frac{i}{10000^{j/d}})]_{j \in [d/2]}$ with $N$ being the number of different tokens and use standard RoPE (Su et al., 2024) as position embedding. We train every model with at least 1M samples to guarantee convergence using Adam with a cosine learning rate. If the model doesn't converge, we retrain using 5M samples. After a grid search over learning rates, we train all the Transformer models with learning rates 1e-3 and the rest of the models with learning rates 3e-4. We run all the experiments on a server with 8 A100s and the estimated time to reproduce the results is within 2 days.

# E    OMITTED PROOF

## E.1    BUILDING BLOCKS OF FFNs CONSTRUCTION

We will first show some basic operations that multiple layers of feedforward neural networks with ReGLU activation can perform that will be used in the following proofs.

**Lemma E.1** (Multiplication). *Given two dimensions $i_1, i_2$, there exists a parameter configuration of a 1-layer feedforward neural network with ReGLU activation that for any input $x \in \mathbb{R}^d$ and constant width, computes the product of $x_{i_1}$ and $x_{i_2}$.*

$$g(x) = [x_{i_1} \times x_{i_2}, 0, \ldots, 0]^\top.$$

*Proof.* We can construct the following feedforward neural network with ReGLU activation:

$$W_1 x = [x_{i_1} \quad -x_{i_1} \quad 0 \ldots \quad 0]^\top,$$
$$W_2 x = [x_{i_2} \quad x_{i_2} \quad 0 \ldots \quad 0]^\top,$$
$$W_3 h = [h_1 + h_2 \quad 0 \quad 0 \quad 0 \quad \ldots \quad 0]^\top,$$
$$b_1 = b_2 = b_3 = 0.$$
$$g(x) = W_3(\text{ReLU}(W_1 x) \otimes \text{ReLU}(W_2 x)) = [x_{i_1} \times x_{i_2}, 0, \ldots, 0]^\top.$$

$\square$

**Lemma E.2** (Linear Operations). *Given a linear transformation $W \in \mathbb{R}^{d \times d}$, there exists a parameter configuration of a 1-layer feedforward neural network with ReGLU activation and width $w = d$ that for any input $x \in \mathbb{R}^d$, computes $Wx$.*

*Proof.*

$$b_1 = 1^w, b_2 = 0, b_3 = 0,$$
$$W_1 = 0, W_2 = W, W_3 = I_{d \times d}$$

$\square$

**Lemma E.3** (Indicator). *Given a constant integer $B \leq d$ and a dimension $i$, there exists a parameter configuration of a 1-layer feedforward neural network with ReGLU activation and width $4$ that for any input $x \in \mathbb{R}^d$, computes the indicator function of $x_i = B$ when $x_i$ is an integer.*

*Proof.*

$b_2 = 1^w, b_1 = [-B - 0.5, -B + 0.5, B - 0.6, B + 0.4]^\top, b_3 = 10,$

$W_2 = 0, W_1 x = [x_i, x_i, -x_i, -x_i]^\top,$

$\text{ReLU}(W_1 x + b_1) = [\text{ReLU}(x_i - B - 0.5), \text{ReLU}(x_i - B + 0.5), \text{ReLU}(B - x_i - 0.6), \text{ReLU}(B - x_i + 0.4)]$

$W_3 = 10 \begin{bmatrix} 1 & -1 & -1 & 1 \end{bmatrix}^\top.$

Then,

$$\begin{aligned}
g(x) =\ & W_3 \mathrm{ReLU}(W_1 x + b_1) + b_3 \\
=\ & 10\left(\mathrm{ReLU}(x_i - B - 0.5) - \mathrm{ReLU}(x_i - B + 0.5)\right) \\
& + 10\left(\mathrm{ReLU}(B - x_i - 0.6) - \mathrm{ReLU}(B - x_i + 0.4)\right) + 10 \\
=\ & \begin{cases}
10 \times 0 + 10 \times -1 + 10 = 0, & \text{if } x_i < B, \\
10 \times -0.5 + 10 \times -0.4 + 10 = 1, & \text{if } x_i = B, \\
10 \times -1 + 10 \times 0 + 10 = 0, & \text{if } x_i > B.
\end{cases}
\end{aligned}$$

$\square$

**Lemma E.4** (Lookup Table). *For constant $B$ and $k$ such that $kB \le d$, given a lookup table which key is tuple of $k$ integers bounded by $B$, and value is a scalar in $\mathbb{R}_p$, there exists a parameter configuration of a $1-$ layer feedforward neural network with ReGLU activation with width $O(B^k)$ that for any input $x \in \mathbb{R}^d$, computes the value of the lookup table at the key $x_{i_1}, x_{i_2}, \ldots, x_{i_k}$.*

*Proof.* We can calculate $x_{i_1} + B \times x_{i_2} + B^2 \times x_{i_3} + \ldots + B^{k-1} \times x_{i_k}$, and then scale $B^k$ indicator functions to get the value of the lookup table at the key $x_{i_1}, x_{i_2}, \ldots, x_{i_k}$. $\square$

**Lemma E.5** (Threshold). *Given any threshold $u$ and constant $\epsilon > 0$, there exists a parameter configuration of a 1-layer feedforward neural network with ReGLU activation and width $2$ that for any input $x \in \mathbb{R}^d$, computes the indicator function $x_i > u$ on $x_i \in [-\infty, u - \epsilon] \cup [u, \infty]$.*

*Proof.*

$$\begin{aligned}
& b_2 = 1^w, b_1 = [-u + \epsilon, -u + 0.5\epsilon]^\top, b_3 = 0, \\
& W_2 = 0, W_1 x = [x_i, x_i]^\top, \\
& \mathrm{ReLU}(W_1 x + b_1) = [\mathrm{ReLU}(x_i - u + \epsilon) \quad \mathrm{ReLU}(x_i - u + 0.5\epsilon)] \\
& W_3 = \frac{2}{\epsilon}[1 \quad -1 \quad -1 \quad 1]^\top.
\end{aligned}$$

Then,

$$\begin{aligned}
g(x) =\ & W_3 \mathrm{ReLU}(W_1 x + b_1) + b_3 \\
=\ & 2/\epsilon\left(\mathrm{ReLU}(x_i - u - \epsilon) - \mathrm{ReLU}(x_i - u + 0.5)\right) \\
=\ & \begin{cases}
0 & \text{if } x_i < u - \epsilon, \\
1, & \text{if } x_i > u,
\end{cases}
\end{aligned}$$

$\square$

**Lemma E.6** (Interval). *Given any constant $u, v$ and $\epsilon > 0$, there exists a parameter configuration of a 1-layer feedforward neural network with ReGLU activation and width $4$ that for any input $x \in \mathbb{R}^d$, computes the indicator function $x_i \in [u, v]$ on $x_i \in [-\infty, u - \epsilon] \cup [u, v] \cup [v, \infty]$.*

*Proof.* The interval function here can be written as the difference of two threshold functions. We can use the same construction as in Lemma E.5 to approximate the indicator function $x_i > u$ and $x_i > v + \epsilon$ and then take the difference. $\square$

### E.2 BUILDING BLOCKS OF TRANSFORMERS CONSTRUCTION

We will show in this section some construction for basic functionality using Transformer Blocks. This construction will be used in the following sections to prove the main results.

We will always use $X \in \mathbb{R}_p^{d \times l}$ as the input to the Transformer Block, where $d$ is the dimension of the input, and $l$ is the length of the sequence. We will first outline all the building functionality and then show how to implement them.

**Definition E.7** (Copying Function). For integer $s$, index set $I_1, I_2 \subset [d-20]$ satisfying $|I_1| = |I_2|$, a copying function $\text{COPY}[s, I_1, I_2]$ satisfies the following, $\forall X \in \mathbb{R}_p^{d \times l}$, then

$$\text{COPY}[s, I_1, I_2](X)_{I_2,k} = x_{I_1, \max\{k-s,0\}} \quad \forall k \leq [m]$$
$$\text{COPY}[s, I_1, I_2](X)_{I_2^c,k} = 0 \quad \forall r \in [m]$$

**Definition E.8** (Counting Function). For index set $I_1, I_2 \subset [d-20], |I_1| = |I_2| \leq 10$ and index $i$, a counting function $\text{COUNT}[I_1, I_2, i]$ satisfies the following, if $\forall v \in I_1 \cup I_2, k \in [l], X_{v,k} \in \mathbb{Z}_p$ and $X_{v,k} \neq 0$, then

$$\text{COUNT}[I_1, I_2, i](X)_{i,k} = \frac{1}{\sum_{h=1}^{k} \mathbf{1}[X_{I_1,h} = X_{I_2,k}] + 1} \quad \forall k \in [l].$$
$$\text{COUNT}[I_1, I_2, i](X)_{i^c,k} = 0 \quad \forall k \in [l].$$

**Definition E.9** (Matching Function). For index set $I_1, I_2, I_3, I_4 \subset [d-20], |I_1| = |I_2| \leq 10, |I_3| = |I_4|$, a matching function $\text{Match}[I_1, I_2, I_3, I_4]$ satisfies the following, if $\forall v \in I_1 \cup I_2, k \in [l], X_{v,k} \in \mathbb{Z}_p$, then

$$\text{Match}[I_1, I_2, I_3, I_4](x)_{I_3,k} = X_{I_4,k^*} \quad \forall k \in [l]$$
$$\text{where } k^* = \begin{cases} \min\{h \mid X_{I_1,h} = X_{I_2,k}\}, \{h \mid X_{I_1,h} = X_{I_2,k}\} \neq \emptyset \\ 1, \text{ otherwise} \end{cases}.$$

**Definition E.10** (Matching Closest Function). For index set $I_1, I_2, I_3, I_4 \subset [d-20], |I_1| = |I_2| \leq 10, |I_3| = |I_4|$, a matching closest function $\text{Match}[I_1, I_2, I_3, I_4]$ satisfies the following, if $\forall v \in I_1 \cup I_2, k \in [l], X_{v,k} \in \mathbb{Z}_p$, then

$$\text{MatchClose}[I_1, I_2, I_3, I_4](x)_{I_3,k} = X_{I_4,k^*} \quad \forall k \in [l]$$
$$\text{where } k^* = \begin{cases} \max\{h \mid X_{I_1,h} = X_{I_2,k}\}, \{h \mid X_{I_1,h} = X_{I_2,k}\} \neq \emptyset \\ 1, \text{ otherwise} \end{cases}.$$

**Definition E.11** (Matching Nearest Function). For index set $I_1, I_2, I_3, I_4 \subset [d-20], |I_1| = |I_2| \leq 10, |I_3| = |I_4|$ and index $i$, a matching nearest function $\text{MatchNearest}[I_1, I_2, I_3, I_4, i]$ satisfies the following, if $\forall v \in I_1 \cup I_2, k \in [l], X_{v,k} \in \mathbb{Z}_p$, then

$$\text{MatchNearest}[I_1, I_2, I_3, I_4](x)_{I_3,k} = X_{I_4,k^*} \quad \forall k \in [l]$$
$$\text{where } k^* = \begin{cases} \arg\min_{h \in \{h|X_{I_1,h}=X_{I_2,k}\}} |h - X_{i,k}|, \{h \mid X_{I_1,h} = X_{I_2,k}\} \neq \emptyset \\ 1, \text{ otherwise} \end{cases}.$$

**Definition E.12** (Matching Next Function). Given any interger constant $A$, assuming $p > 10A \log n$, for index set $I_1, I_2, I_3, I_4 \subset [d-20], |I_1| = |I_2| \leq 10, |I_3| = |I_4|$, and a special counting index $a$, a matching next function $\text{MatchNext}[I_1, I_2, I_3, I_4, a]$ satisfies the following, if $X$ satisfies the following condition:

1. $\forall v \in I_1 \cup I_2, k \in [l], X_{v,k} \in \mathbb{Z}_p$,

2. $X_{a,k} \in \text{ROUND}(1/[n^A], p) \cup \{0\}$,

3. For any $k \in [l]$, given any $k \geq k$, the counting index multiset $S_k = \{X_{a,k'} \mid X_{I_1,k'} = X_{I_2,k}\}$ takes consecutive and disjoint values in $\text{ROUND}(1/[n^A], p)$, that is, there exists $u_k, v_k \in \text{ROUND}(1/[n^A], p)$ such that $S_k = [u_k, v_k] \cap \text{ROUND}(1/[n^A], p)$.

then, we have

$$\text{MatchNext}[I_1, I_2, I_3, I_4, a](X)_{I_3,k} = X_{I_4,k^*} \quad \forall k \in [l]$$
$$\text{where } k^* = \arg\min_{h \in \{h|X_{I_1,h}=X_{I_2,k}\} \cup \{1\}} |X_{a,h} - \text{next}(X_{a,k})|.$$

Now we will show how to implement these functions using Transformer Blocks. The construction here is motivated by the construction in Feng et al. (2023) with some modifications.

**Lemma E.13** (Copying Blocks). *For integer $s$, index set $I_1, I_2 \subset [d-10]$ satisfying $|I_1| = |I_2|$, a copying function $\text{COPY}[s, I_1, I_2]$ can be implemented with 1 feedforward block $g$ and 1 attention block $\mathcal{A}$ with 1 attention head. Formally, when $X_{d,k} = k$, it holds that*

$$\mathcal{A}\left(g\left(X\right) + X\right) = \text{COPY}[s, I_1, I_2](X).$$

*Proof of Lemma E.13.* We will use the feedforward block to calculate $X_{k,d}^2$ and 1 (Lemma E.1) and have

$$(g(X) + X)_{d-1,k} = k^2$$
$$(g(X) + X)_{d-2,k} = 1.$$
$$\forall i \notin \{d-1, d-2\}, (g(X) + X)_{i,k} = X_{i,k}.$$

We will use $X'$ to denote $g(X) + X$. Then we will choose $W^{(K)}, W^{(Q)}$ such that

$$W^{(K)} X'_{:,k'} = n \begin{bmatrix} 1 \\ k' \\ k'^2 \end{bmatrix}$$

$$W^{(Q)} X'_{:,k} = \begin{bmatrix} -(k^2 + s^2 - 2sk) \\ 2k - 2s \\ -1 \end{bmatrix}$$

Hence

$$\left(\left(W^{(K)} X\right)^\top \left(W^{(Q)} X\right)\right)_{k',k}$$
$$= -n\left(k'^2 - k'(2k - 2s) + k^2 + s^2 - 2sk\right)$$
$$= -n(k - s - k')^2$$

Hence we have

$$\arg\max_{k' < k} \left(\left(W^{(K)} X\right)^\top \left(W^{(Q)} X\right)\right)_{k',k} = \max\{k - s, 0\}.$$

Also, for any $k' \leq k, k' \neq \max\{k - s, 0\}$, we have

$$\left(\left(W^{(K)} X\right)^\top \left(W^{(Q)} X\right)\right)_{k',k} - \left(\left(W^{(K)} X\right)^\top \left(W^{(Q)} X\right)\right)_{\max\{k-s,0\},k} < -n.$$

Hence after the column-wise softmax and rounding to $p = O(\log n)$ bit, we have

$$\left(\text{softmax}\left(\left(W^{(K)} X\right)^\top \left(W^{(Q)} X\right)\right)\right)_{k',k} = \mathbf{1}[k' = \max\{k - s, 0\}].$$

We will then choose $W^{(V)}$ such that

$$W^{(V)} X'_{I_2,k'} = X'_{I_1,k'} = X_{I_1,k'} \quad \forall k' \in [l].$$
$$W^{(V)} X'_{I_2^c,k'} = 0 \quad \forall k' \in [l].$$

This then concludes that

$$\mathcal{A}\left(g\left(X\right) + X\right) = \text{COPY}[s, I_1, I_2](X).$$

The proof is complete. □

**Lemma E.14** (Counting Blocks). *For index set $I \subset [d-20]$ satisfying $|I_1| = |I_2| \leq 10$, a counting function $\text{COUNT}[i, I_1, I_2]$ can be approximated with 1 feedforward block $g$ and 1 attention block $\mathcal{A}$ with 1 attention head. Formally, when $X_{d,k} = k$ and $X_{3,k} = \mathbf{1}[k = 1], X_{I_1,1} = 0$, it holds that*

$$\mathcal{A}\left(g\left(X\right) + X\right)_{i,k} = \text{ROUND}\left(\text{COUNT}[s, I_1, I_2](X)_{i,k}, p\right).$$
$$\mathcal{A}\left(g\left(X\right) + X\right)_{i^c,k} = 0.$$

*Proof of Lemma E.14.* We will use the feedforward block to calculate $X^2_{v,k}, v \in I_1 \cup I_2$ (Lemma E.1) and have

$$(g(X) + X)_{d-i,k} = X^2_{I_1[i],k}, i \in [|I|].$$
$$(g(X) + X)_{d-|I|-i,k} = X^2_{I_2[i],k}, i \in [|I|].$$
$$(g(X) + X)_{d-2|I|-1,k} = 1.$$
$$\forall i \notin \{d - i \mid i \in [2|I| + 1]\}, (g(X) + X)_{i,k} = X_{i,k}.$$

We will use $X'$ to denote $g(X) + X$. Then we will choose $W^{(K)}, W^{(Q)}$ such that

$$W^{(K)} X'_{:,k'} = n \begin{bmatrix} 1 + \mathbf{1}[k' = 1] \\ X_{I_1[i],k'} \\ X^2_{I_1[i],k'} \end{bmatrix}_{i \in [I]}$$

$$W^{(Q)} X'_{:,k} = \begin{bmatrix} X^2_{I_2[i],k} \\ -X_{I_2[i],k} \\ 1 \end{bmatrix}_{i \in [I]}$$

Hence,

$$\left( \left( W^{(K)} X \right)^\top \left( W^{(Q)} X \right) \right)_{k',k}$$

$$= -n \sum_{i=1}^{|I|} \left( X'^2_{I_2[i],k'} - X_{I_1[i],k'}(2X_{I_2[i],k}) + X^2_{I_2[i],k} \right) + n\mathbf{1}[k' = 1] \sum_{i=1}^{|I|} X^2_{I[i],k}.$$

$$= -n \sum_{i=1}^{|I|} (X_{I_1[i],k'} - X_{I_2[i],k})^2 + n\mathbf{1}[k' = 1] \sum_{i=1}^{|I|} X^2_{I_2[i],k}.$$

Hence we have

$$\max_{k' < k} \left( \left( W^{(K)} X \right)^\top \left( W^{(Q)} X \right) \right)_{k',k} = 0.$$

Equality holds when $k' = 1$ or $X_{I_1[i],k'} = X_{I_2[i],k}$ for all $i \in [|I_1|]$.

Also, for any $k' \leq k, k' \neq 1$ or $X_{I_1[i],k'} \neq X_{I_2[i],k}$ for some $i \in [|I_1|]$, we have

$$\left( \left( W^{(K)} X \right)^\top \left( W^{(Q)} X \right) \right)_{k',k} < -n.$$

Hence after the column-wise softmax and rounding to $p = O(\log n)$ bit, we have

$$\left( \text{softmax} \left( \left( W^{(K)} X \right)^\top \left( W^{(Q)} X \right) \right) \right)_{k',k} = \text{ROUND} \left( \frac{1}{\sum_{h=1}^k \mathbf{1}[X_{I_1,h} = X_{I_2,k}] + 1}, p \right)$$

Here the $O\left(\frac{1}{n^A}\right)$ term comes from the fact that the softmax is rounded to $p = O(\log n)$ bit.

We will then choose $W^{(V)}$ such that

$$W^{(V)} X'_{i,k'} = X'_{3,k'} = \mathbf{1}[k' = 1] \quad \forall k' \in [l].$$
$$W^{(V)} X'_{I^c,k'} = 0 \quad \forall k' \in [l].$$

This then concludes that

$$\mathcal{A} \left( g\left( X \right) + X \right)_{i,k} = \text{ROUND} \left( \text{COUNT}[s, I_1, I_2](X)_{i,k}, p \right).$$
$$\mathcal{A} \left( g\left( X \right) + X \right)_{i^c,k} = 0.$$

$\square$

**Lemma E.15** (Matching Blocks). *Given any constant $c$, for index set $I_1, I_2, I_3, I_4 \subset [d-20], |I_1| = |I_2| \le 10, |I_3| = |I_4|$, a matching function $\text{Match}[I_1, I_2, I_3, I_4]$ can be implemented with 1 feedforward block $g$ and 1 attention block $\mathcal{A}$ with 1 attention head. Formally, when $X_{d,k} = k, X_{3,k} = \mathbf{1}[k=1], X_{I_1,1} = 0$ and $k \le n^c$, it holds that*

$$\mathcal{A}(g(X) + X) = \text{Match}[I_1, I_2, I_3, I_4](X)$$

*Proof.* We will use the feedforward block to calculate $k^2, X_{v,d}^2, v \in \cup I_1 \cup I_2$ as in the proof of Lemmas E.13 and E.14.

We then choose $W^{(K)}, W^{(Q)}$ such that

$$\left(\left(W^{(K)}X\right)^\top \left(W^{(Q)}X\right)\right)_{k',k}$$

$$= -n^{4c+1}\sum_{i=1}^{|I|}(X_{I_1[i],k'} - X_{I_2[i],k})^2 - nk'^2$$

$$+ \mathbf{1}[k'=1]\left(n^{4c+1}\sum_{i=1}^{|I|}X_{I_2[i],k}^2 + n - n^{2c+2}\right).$$

The detailed construction of $W^{(K)}, W^{(Q)}$ is omitted here since it is similar to the proof of Lemmas E.13 and E.14.

We will discuss several cases for the distribution of $\left(\left(W^{(K)}X\right)^\top \left(W^{(Q)}X\right)\right)_{k',k}$. It always holds that $\left(\left(W^{(K)}X\right)^\top \left(W^{(Q)}X\right)\right)_{1,k} = -n^{2c+2}$.

1. If there doesn't exists $k'$, such that $X_{k',I_1} = X_{k,I_2}$, then for any $i > 1$, we have $\left(\left(W^{(K)}X\right)^\top \left(W^{(Q)}X\right)\right)_{i,k} < -n^{4c+1}$.

2. If there exists $k'$, such that $X_{k',I_1} = X_{k,I_2}$, then for such $k'$, we have $\left(\left(W^{(K)}X\right)^\top \left(W^{(Q)}X\right)\right)_{k',k} = -nk'^2 > -n^{2c+1}$. The rest of the entries are all smaller than $-n^{4c+1}$. Finally, the largest $k'$ satisfying that $X_{k',I_1} = X_{k,I_2}$ will correspond to a $\left(\left(W^{(K)}X\right)^\top \left(W^{(Q)}X\right)\right)_{k',k}$ that is at least $n$ larger than the second largest $\left(\left(W^{(K)}X\right)^\top \left(W^{(Q)}X\right)\right)_{k',k}$, as in the proof of Lemma E.13.

Concluding the above discussion, we have after the column-wise softmax and rounding to $p = O(\log n)$ bit,

$$\left(\text{softmax}\left(\left(W^{(K)}X\right)^\top \left(W^{(Q)}X\right)\right)\right)_{k',k} = \begin{cases} \mathbf{1}[k' = \min\{h \mid X_{I_1,h} = X_{I_2,k}\}], \{h \mid X_{I_1,h} = X_{I_2,k}\} \ne \emptyset \\ \mathbf{1}[k'=1], \text{otherwise} \end{cases}$$

Further, we will choose $W^{(V)}$ such that

$$W^{(V)}X'_{I_3,k'} = X'_{I_4,k'} = X_{I_4,k'} \quad \forall k' \in [l].$$
$$W^{(V)}X'_{I_3^c,k'} = 0 \quad \forall k' \in [l].$$

This then concludes that

$$\mathcal{A}(g(X) + X) = \text{Match}[I_1, I_2, I_3, I_4](X)$$

This concludes the proof. $\square$

**Lemma E.16** (Matching Closest Blocks). *Given any constant c, for index set $I_1, I_2, I_3, I_4 \subset [d-20], |I_1| = |I_2| \leq 10, |I_3| = |I_4|$, a matching closest function $\mathrm{MatchClose}[I_1, I_2, I_3, I_4]$ can be implemented with 1 feedforward block g and 1 attention block $\mathcal{A}$ with 1 attention head. Formally, when $X_{d,k} = k, X_{3,k} = \mathbf{1}[k=1], X_{I_1,1} = 0$ and $k \leq n^c$, it holds that*

$$\mathcal{A}\left(g\left(X\right) + X\right) = \mathrm{MatchClose}[I_1, I_2, I_3, I_4](X)$$

*Proof.* The proof is similar to the proof of Lemma E.15, and one can design the attention pattern as

$$\left(\left(W^{(K)}X\right)^{\top}\left(W^{(Q)}X\right)\right)_{k',k}$$

$$= -n^{4c+1}\sum_{i=1}^{|I|}(X_{I_1[i],k'} - X_{I_2[i],k})^2 - n(k-k')^2$$

$$+ \mathbf{1}[k'=1]\left(n^{4c+1}\sum_{i=1}^{|I|}X_{I_2[i],k}^2 + n(k-1)^2 - n^{2c+2}\right).$$

The rest of the proof is omitted here. $\qquad\square$

**Lemma E.17** (Matching Nearest Blocks). *Given any constant c, for index set $I_1, I_2, I_3, I_4 \subset [d-20], |I_1| = |I_2| \leq 10, |I_3| = |I_4|$ and index i , a matching nearest function $\mathrm{MatchNearest}[I_1, I_2, I_3, I_4, i]$ can be implemented with 1 feedforward block g and 1 attention block $\mathcal{A}$ with 1 attention head. Formally, when $X_{d,k} = k, X_{3,k} = \mathbf{1}[k=1], X_{I_1,1} = 0$ and $k \leq n^c$, it holds that*

$$\mathcal{A}\left(g\left(X\right) + X\right) = \mathrm{MatchNearest}[I_1, I_2, I_3, I_4, i](X)$$

*Proof.* The proof is similar to the proof of Lemma E.15, and one can design the attention pattern as

$$\left(\left(W^{(K)}X\right)^{\top}\left(W^{(Q)}X\right)\right)_{k',k}$$

$$= -n^{4c+1}\sum_{u=1}^{|I|}(X_{I_1[u],k'} - X_{I_2[u],k})^2 - n(X_{i,k} - k')^2$$

$$+ \mathbf{1}[k'=1]\left(n^{4c+1}\sum_{u=1}^{|I|}X_{I_2[u],k}^2 + n(1 - X_{i,k})^2 - n^{2c+2}\right).$$

The rest of the proof is omitted here. $\qquad\square$

**Lemma E.18** (Matching Next Blocks). *Given any constant A, c, for index set $I_1, I_2, I_3, I_4 \subset [d-20], |I_1| = |I_2| \leq 10, |I_3| = |I_4|$ and a special counting index a, a matching next function $\mathrm{MatchNext}[I_1, I_2, I_3, I_4, a]$ can implement with 1 feedforward block g and 1 attention block $\mathcal{A}$ with 1 attention head. Formally, when $X_{d,k} = k, X_{3,k} = \mathbf{1}[k=1], X_{I_1,1} = 0$ and $k \leq n^c$, it holds that*

$$\mathcal{A}\left(g\left(X\right) + X\right) = \mathrm{MatchNext}[I_1, I_2, I_3, I_4, a](X)$$

*Proof.* We will use the feedforward block to calculate the following $\overline{\mathrm{next}}$ function, where

$$\overline{\mathrm{next}}(x) = \begin{cases} \frac{1}{2}, & x \geq \frac{2}{3}. \\ \frac{1}{3}, & \frac{3}{5} > x > \frac{2}{5}. \\ \frac{1}{4}, & \frac{7}{20} > x > \frac{3}{10} \\ x - x^2 + x^3, & x \leq \frac{11}{40}. \end{cases}$$

The value can be arbitrary for $x \in [\frac{11}{40}, \frac{3}{10}] \cup [\frac{2}{5}, \frac{7}{20}] \cup [\frac{3}{5}, \frac{2}{3}]$. This function is achievable by a feedforward block through combination of Lemmas E.1 and E.6.

The purpose of this is to approximate the next function for $x \in \mathrm{ROUND}(1/[n^A], p)$, and we have the following lemma.

**Lemma E.19.** *For large enough $n$ and any $x \in \mathrm{ROUND}(1/[n^A], p)$, we have*

$$|\overline{\mathrm{next}}(x) - \mathrm{next}(x)| \le \mathrm{next}(x)^3 + O\left(\frac{1}{n^{10A}}\right).$$

*Proof.* We always have $\mathrm{ROUND}(1/[n^A], p) \cap \left([\frac{11}{40}, \frac{3}{10}] \cup [\frac{2}{5}, \frac{7}{20}] \cup [\frac{3}{5}, \frac{2}{3}]\right) = \emptyset$. We will discuss several cases for $x \in \mathrm{ROUND}(1/[n^A], p)$.

1. If $x \ge \frac{3}{10}$, then $\overline{\mathrm{next}}(x) = \mathrm{next}(x)$.

2. If $x \le \frac{7}{20}$, it holds that $|x - 1/m| \le 1/n^{10A}, m \ge 3$, then

$$\overline{\mathrm{next}}(x) = x - x^2 = \frac{1}{m} - \frac{1}{m^2} + \frac{1}{m^3} + O\left(\frac{1}{n^{10A}}\right)$$

$$= \frac{1}{m+1} - \frac{1}{m^3(m+1)} + O\left(\frac{1}{n^{10A}}\right)$$

This then concludes the proof. $\qquad\square$

We then choose $W^{(K)}, W^{(Q)}$ such that

$$\left(\left(W^{(K)}X\right)^\top \left(W^{(Q)}X\right)\right)_{k',k}$$

$$= -n^{4A+3} \sum_{i=1}^{|I|} (X_{I_1[i],k'} - X_{I_2[i],k})^2 - n^{4A+1}(\overline{\mathrm{next}}(X_{a,k}) - X_{a,k'})^2$$

$$+ \mathbf{1}[k'=1]\left(n^{4A+3} \sum_{i=1}^{|I|} X_{I_2[i],k}^2 + n^{4A+1} X_{a,k}^2 - n^{4A+2}\right).$$

Again, the detailed construction of $W^{(K)}, W^{(Q)}$ is omitted here since it is similar to the proof of Lemmas E.13 and E.14.

We will discuss several cases for the distribution of $\left(\left(W^{(K)}X\right)^\top \left(W^{(Q)}X\right)\right)_{k',k}$. It always holds that $\left(\left(W^{(K)}X\right)^\top \left(W^{(Q)}X\right)\right)_{1,k} = -n^{4A+2}$.

1. If there doesn't exists $k'$, such that $X_{k',I_1} = X_{k,I_2}$, then for any $i > 1$, we have $\left(\left(W^{(K)}X\right)^\top \left(W^{(Q)}X\right)\right)_{i,k} < -n^{4A+3}$.

2. If there exists $k'$, such that $X_{k',I_1} = X_{k,I_2}$, then for such $k'$, we have $\left(\left(W^{(K)}X\right)^\top \left(W^{(Q)}X\right)\right)_{k',k} = -n^{3A}(\mathrm{next}(X_{a,k}) - X_{a,k'})^2 > -n^{4A+1}$. The rest of the entries are all smaller than $-n^{4A+2}$.

   It remains to discuss the distribution of $\left(\left(W^{(K)}X\right)^\top \left(W^{(Q)}X\right)\right)_{k',k}$ for $k'$ satisfying $X_{k',I_1} = X_{k,I_2}$. When $X$ satisfies the condition in Definition E.12, we have that $S_k = \{X_{a,k'} \mid X_{k',I_1} = X_{k,I_2}\}$ takes consecutive and disjoint values in $\mathrm{ROUND}(1/[n^A], p)$. Hence, if $|S_k| > 2$, suppose $y, z \in S_k$ satisfies that

$$|y - \overline{\mathrm{next}}(X_{a,k})| = \min_{x \in S_k} |x - \overline{\mathrm{next}}(X_{a,k})|$$

$$|z - \overline{\mathrm{next}}(X_{a,k})| = \min_{x \in S_k, x \ne y} |x - \overline{\mathrm{next}}(X_{a,k})|.$$

   We will discuss several cases for $y, z$.

- If $y - \overline{\text{next}}(X_{a,k})$ and $z - \overline{\text{next}}(X_{a,k})$ are both negative, then $y > z$, we have,

$$\left(y - \overline{\text{next}}(X_{a,k})\right)^2 - \left(z - \overline{\text{next}}(X_{a,k})\right)^2 = (y-z)(y+z-2\overline{\text{next}}(X_{a,k}))$$
$$\leq -(y-z)^2 \leq -\frac{1}{4n^{4A}}.$$

.
- If $y - \overline{\text{next}}(X_{a,k})$ and $z - \overline{\text{next}}(X_{a,k})$ are both positive, then $y < z$, and same as above we have

$$\left(y - \overline{\text{next}}(X_{a,k})\right)^2 - \left(z - \overline{\text{next}}(X_{a,k})\right)^2 = (y-z)(y+z-2\overline{\text{next}}(X_{a,k}))$$
$$\leq -(y-z)^2 \leq -\frac{1}{4n^{4A}}.$$

.
- If $y - \overline{\text{next}}(X_{a,k})$ and $z - \overline{\text{next}}(X_{a,k})$ have different signs, then according to Lemma E.19, we have, $y = \text{ROUND}(\text{next}(X_{a,k}), p)$ because $S_k$ takes consecutive and disjoint values in $\text{ROUND}(1/[n^A], p)$. This then implies that

$$\left(y - \overline{\text{next}}(X_{a,k})\right)^2 - \left(z - \overline{\text{next}}(X_{a,k})\right)^2$$
$$\leq O(\frac{1}{n^{10A}}) + \frac{1}{\text{next}^6(X_{a,k})} - \left(\frac{1}{\text{next}(X_{a,k})(\text{next}(X_{a,k})+1)}\right)^2$$
$$\leq -\frac{1}{4n^{4A}}.$$

Concluding, we always have for any $k'' \neq k^* = \arg\max_{k',k}\left(\left(W^{(K)}X\right)^\top\left(W^{(Q)}X\right)\right)_{k',k}$

$$\left(\left(W^{(K)}X\right)^\top\left(W^{(Q)}X\right)\right)_{k',k} - \left(\left(W^{(K)}X\right)^\top\left(W^{(Q)}X\right)\right)_{k^*,k} \leq -\frac{n}{4}.$$

Concluding the above discussion, we have after the column-wise softmax and rounding to $p = O(\log n)$ bit,

$$\left(\text{softmax}\left(\left(W^{(K)}X\right)^\top\left(W^{(Q)}X\right)\right)\right)_{k',k} = \mathbf{1}\left[k' = \arg\min_{h\in\{h|X_{I_1,h}=X_{I_2,k}\}\cup\{1\}}|X_{a,h} - \text{next}(X_{a,k})|\right].$$

Further, we will choose $W^{(V)}$ such that

$$W^{(V)}X'_{I_3,k'} = X'_{I_4,k'} = X_{I_4,k'} \quad \forall k' \in [l].$$
$$W^{(V)}X'_{I_3^c,k'} = 0 \quad \forall k' \in [l].$$

This then concludes the proof. $\qquad\square$

### E.3 BUILDING BLOCKS OF RNNs CONSTRUCTION

We will now describe the building blocks of Linear RNNs construction. We will introduce some basic operations that will be used to build more complex RNNs family.

**Lemma E.20** (Recent Input Memorizing). *Given any constant $k$ and $C$, there exists a parameter configuration of linear unit that maintains $C$ dimensions of last $k$ input vectors in the state.*

*Proof.* Suppose the input sequence is $x_{1:t} \in \mathbb{R}^d$, and the dimensions that the state should memorize are $d_1, d_2, \ldots, d_C$. We can construct the following linear unit:

$$h_t = \begin{bmatrix} x_{t-1,d_1} & \cdots & x_{t-1,d_C} & h_{t-1,1} & \cdots & h_{t-1,C\times(k-1)} & h_{t-1,C\times k+1} & \cdots & h_{t-1,d} \end{bmatrix}.$$

$\qquad\square$

**Lemma E.21** (Summation). *Given any constant $k$ and $C$, there exists a parameter configuration of linear unit that maintains the sum of one dimension of the last $k$ input vectors in the state.*

*Proof.* Suppose WLOG the input sequence is $x_{1:t} \in \mathbb{R}^d$, and the dimension that the state should memorize is 1. We can construct the following linear unit:

$$h_t = [x_{t-1,1} + h_{t-1,1} \quad h_{t-1,2} \dots \quad h_{t-1,d}].$$

$\square$

**Lemma E.22** (Special Position Memorizing). *Given any constant $k$ and $C$, there exists a parameter configuration of linear unit and a FFN with $Ck$ width that maintains the $C$ dimensions of the input vector at position 1 to $k$ in the state.*

*Proof.* This is a direct combination of Lemma E.21 and Lemmas E.1 and E.4. The FFN can assign all the input vectors with position greater than $k$ to 0, and permute the corresponding dimensions of first $k$ input vectors to the first $Ck$ dimensions of the state. The linear unit can then maintain the state. $\square$

**Lemma E.23** (Recite Fixed Sequence). *Given any constant integer $k$ and $C$, there exists a FFN with width $kC$ that can output fixed sequence of scalars that takes values in $[C]$ on a fixed set of positions $l_1, \dots l_k$.*

*Proof.* This is a direct consequence of Lemma E.4. $\square$

### E.4 PROOF OF THEOREM 4.6

We will first restate the theorem for clarity.

**Theorem 4.6.** *For task $T \in \{$Index, AR, c-gram retrieval, Counting$\}$, there exist constant-size Transformers that can solve $T$. On the other hand, any RNN with $o(n)$-bit memory cannot solve $T$ of size $n$ with any length of CoT for large enough $n$.*

*Proof.* We will discuss by cases.

When $T$ is Index, we will first show why RNN cannot solve the Index question without $\Omega(n)$ memory. The key observation is that the recurrent form of RNNs allowed the algorithm to be run in a streaming fashion with $o(n)$ bit memory. Here streaming means that the algorithm gets to look at each bit of the memory sequentially and can only update a constant size of memory.

**Lemma E.24.** *Consider the following two-party game, where Alice receives string $x \in \{0, 1\}^n$ and Bob receives an integer $k$, and Bob wants to know the value of $x_k$. If only Alice is allowed to send a signal to Bob, then $\Omega(n)$ bit communication complexity is required.*

*Proof of Lemma E.24.* Suppose there exists a communication protocol where $B$ only receives $o(n)$ bit and can perfectly decide $x_k$. Because Alice doesn't know $k$, the protocol must send the same message to Bob for all $k$. Hence Bob can reconstruct the whole string $x$ with $n$ bit with $o(n)$ bit communication. This is a contradiction. $\square$

Now if RNN can solve the Index problem with $o(n)$ bit memory, then it can also solve the Index problem with $o(n)$ bit communication complexity. This is because Alice can simply run the RNN on input $x$ and send the hidden state to Bob. Then Bob can run the RNN with the hidden state and $k$ to get the answer. This is a contradiction to Lemma E.24. Hence RNN cannot solve the Index problem with $o(n)$ bit memory.

On the other hand, we will show that Transformers can solve the Index problem with $O(\log n)$ bit parameters. This is because using 2 layers of Transformer, we will implement a Match Block (Lemma E.15) that can match the last query token with the position of the previous token and retrieve the type of the matched token to the query token.

When $T$ is AR, wthout loss of generality, we assume that $n$ is even. The proof is similar to the proof of the proof of theIndex problem. As there are $n$ different types of tokens, we can label them as $[n]$. Now for any boolean sequence $x \in \{0, 1\}^{n/2}$, solving AR for the following input is equivalent to solving the Index problem for $x$:

$$\mathcal{S}_{\text{in}} = \text{}, 1, x_1 + n/2, 2, x_2 + n/2, \dots, n/2, x_{n/2} + n/2, k$$

This then implies that RNN cannot solve AR with $o(n)$ bit memory. Transformers, on the other hand, can still solve AR with $O(\log n)$ bit parameters, we will use one layer of copying function to copy each token's previous token's type to it. Then we can use the Match Block to match the last query token with the position of the previous token and retrieve the type of the matched token to the query token.

When $T$ is $c$-gram retrieval, without loss of generality, we assume that $n$ is a multiple of $c$. The proof is similar to the proof of Theorem 4.6. As there are $n$ different types of tokens, we can label them as $[n]$. Now for any boolean sequence $x \in \{0, 1\}^{n/2}$, solving AR for the following input is equivalent to solving the Index problem for $x$:

$$\mathcal{S}_{\text{in}} = \texttt{}, \underbrace{1, \ldots, 1}_{c-1}, x_1 + n/c, \underbrace{2, \ldots, 2}_{c-1}, x_2 + n/c, \ldots, \underbrace{n/c, \ldots, n/c}_{c-1}, x_{n/c} + n/c, \underbrace{k, \ldots, k}_{c-1}$$

This then implies that RNN cannot solve $c$-gram retrieval with $o(n)$ bit memory. Transformers, on the other hand, can still solve $c$-gram retrieval with $O(\log n)$ bit parameters, we will use one layer of copying function to copy each token's previous $c - 1$ tokens' type to it. Then we can use the Match Block to match the last query token with the position of the previous token and retrieve the type of the matched token to the query token.

When $T$ is Counting, we will first show why RNN cannot solve the Counting question without $\Omega(n)$ memory. Consider the following setup, given any $x \in \{0, 1\}^n$, the input string is $j_1 j_2 \ldots j_k$ where $\{j_i \ldots j_k\} = \{j \mid x_j = 1\}$, then solving the Counting question for this input string for queried threshold 1 is equivalent to solving the Index problem for $x$. This then implies that RNN cannot solve the Counting question with $o(n)$ bit memory.

On the other hand, we will show that Transformers can solve the Counting question with $O(\log n)$ bit parameters. This is because using 2 layers of Transformer, we can first use a COPY block to copy the last query token to the token corresponds to the threshold, and then use a COUNT block (Lemma E.14) that can count the number $m$ of the appearance of the last query token in the input sequence, and then write $1/(m + 1)$ to one of the dimension. Finally, we can use the Feed Forward Network on the last layer to multiply threshold $+1$ with this value and compare the result to 1 to get the answer. $\square$

### E.5 Proof of Theorem 4.1

We will first prove a lemma assuming PSPACE $\not\subset$ P/poly.

**Lemma E.25.** *If* PSPACE $\not\subset$ P/poly*, then there exists a Turing machine $M$ with linear space complexity that cannot be simulated by a polynomial-size circuit family.*

*Proof.* We will prove this by contradiction. Assuming for every Turing machine $M$ with linear space complexity, there exists a polynomial-size circuit family $\{C_n\}$ that can simulate $M$. We will construct a polynomial-size circuit family $\{C'_n\}$ that can decide PSPACE, which contradicts the assumption that PSPACE $\not\subset$ P/poly.

Given any language $L \in$ PSPACE, we can decide $L$ by a Turing machine $M_L$ with space $O(n^k)$ for some constant $k$. We can consider another language $L' = \{x1^{|x|^k} \mid x \in L\}$. We can decide $L'$ by a Turing machine $M_{L'}$ with linear space complexity by checking the length of the input and then simulating $M_L$. By the assumption, there exists a polynomial-size circuit family $\{C_n\}$ that can simulate $M_{L'}$. We can then construct a polynomial-size circuit family $\{C'_n\}$ that can decide $L$ by first counting the length of the input and then simulating $C_n$ on the extended input. This contradicts the assumption that PSPACE $\not\subset$ P/poly. $\square$

Now we are ready to prove our theorem.

**Theorem 4.1.** *Assuming* PSPACE $\not\subset$ P/poly*, there exists an algorithmic problem such that (1) there exist constant-size Linear RNNs that can solve the problem with polynomial length CoT; and (2) any constant-size regular RNNs cannot solve the problem without CoT.*

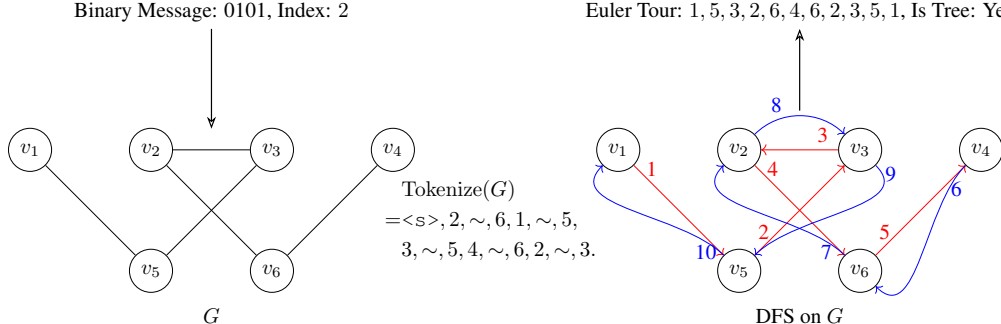

Figure 6: An example of the graph constructed from the binary sequence $x = 0101$ and the index $k = 2$ and the corresponding DFS tour.

*Proof.* By Lemma E.25, we know that if PSPACE $\not\subset$ P/poly, then there exists a Turing machine $M$ with linear space complexity that cannot be simulated by a polynomial-size circuit family. We will use this result to prove Theorem 4.1.

We design the task as follows, for any $n$, let $m = \lfloor \log_2 n \rfloor$, for any boolean input $x$ of length $m$, we will choose input sequence as $\mathcal{S}_{\text{in}} = \overline{0^{n-m}x}$ and the label as $y = $ YES if $M(x) = 1$ and $y = $ NO otherwise.

Because we are considering regular RNN with $O(m)$ memory, we know that we can compute the result of RNN without CoT through a circuit family with size $\text{Poly}(m)$. However, we know that $M$ cannot be simulated by a polynomial-size circuit family. Hence, no RNN family with $O(m)$ memory can solve the task for all $n$.

On the other hand, we can simulate $M$ by the RNN by maintaining the state, the pointer, and the tape of the $M$ inside the state of the RNN. The RNN can then maintain the transition function of the Turing machine in its output function as a lookup table Lemma E.4 and write down the updated state, the direction of the pointer movement, and the updated cell value at the pointer in its context. Paired with the ability to memorize the recent input Lemma E.20, the RNN can then simulate the running of the Turing machine.

Because the space complexity of $M$ is linear in $m$, the time complexity of $M$ is $\exp(O(m))$ which is polynomial in $n$. Hence, we can solve the task by an RNN with CoT and $O(m)$ memory and polynomial-size circuit family. $\qquad\square$

### E.6   PROOF OF THEOREM 4.7

We will now proceed to prove our main theorem, which states that Transformers with chain-of-thought can solve IsTree perfectly, while RNNs cannot. We will first restate the theorem here.

**Theorem 4.7.** *There exist constant-size Transformers that can solve IsTree with CoT of length $O(n)$. On the other hand, any RNN with $o(n)$-bit memory cannot solve IsTree with any length of CoT.*

*Proof of Theorem 4.7.* We will prove this theorem by proving the following lemmas.

**Lemma E.26.** *For any $n$ and RNN $R$ with $o(n)$ memory, $R$ cannot perfectly solve IsTree of size $n$.*

**Lemma E.27.** *There exists a Transformer $T$ with constant depth and width, and $O(\log n)$ precision, that can solve IsTree of size $n$ perfectly with Chain of Thought.*

This proof is a direct combination of Lemmas E.26 and E.27. $\qquad\square$

### E.6.1   PROOF OF LEMMA E.26

We first reduce another problem in communication complexity to IsTree.

**Lemma E.28.** *Consider the following two-party game, where Alice receives string $x \in \{0,1\}^n$ and Bob receives an integer $k$, and Bob wants to know whether $x_k = x_{k-1}$. If only Alice is allowed to send information, $\Omega(n)$ bit communication complexity is required.*

*Proof of Lemma E.28.* We can reduce this problem to the problem in Lemma E.24. Considering the game in Lemma E.24, given any $x \in \{0,1\}^n$, we can construct $\tilde{x}_i = \sum_{j=1}^{i} x_i \mod 2$. Then $\tilde{x}$ is a string of length $n$ with only 0 and 1. Moreover, $x_k = x_{k-1}$ if and only if $\tilde{x}_k = \tilde{x}_{k-1}$. Hence, if Bob can solve the problem in Lemma E.28 with $o(n)$ bit, he can solve the problem in Lemma E.24. This is a contradiction. □

*Proof of Lemma E.26.* Now suppose that we have a streaming algorithm for IsTree with only $o(n)$ memory. We shall prove Alice and Bob in Lemma E.28 can use it to solve the original question with $o(n)$ memory.

Consider the following graph with $n+2$ nodes. There is a node $i$ corresponding to each $x_i$ for $i \in [n]$ and two special nodes $n+1, n+2$. Node $i$ will be connected to $n+1$ if $x_i = 0$ and to $n+2$ if $x_i = 1$. Moreover, $k-1$ and $k$ will be connected. Now the original answer Bob wants is False if and only if the graph is a Tree. Hence, given access to the streaming algorithm, Alice can run it on the edges that she knows exist and send the memory to Bob. Bob can then run it on the edges that he knows exist. Combining they will be able to solve the original problem. This is a contradiction.

Moreover, as RNN with CoT is also a streaming algorithm, it also requires $\Omega(n)$ memory. □

E.6.2 PROOF OF LEMMA E.27

*Proof of Lemma E.27.* The proof is two-folded. We will first define an algorithm that can solve IsTree by generating a sequence of vertices of length $O(n)$, and then we will show that this sequence can be generated by a Transformer with constant depth and width, and $O(\log n)$ precision as a Chain of Thought.

---

**Algorithm 1** Depth-First Search Algorithm

**Require:** A graph $G = (V, E)$ with $n$ vertices and $E$ has an ordering $e_1, \ldots, e_m$.
1: Initialize two stacks of vertices $S_1, S_2$ with $S_1 = [v_1], S_2 = \emptyset$.
2: **while** $S_1$ is not empty **do**
3:     Let $v$ be the top of $S_1$. Yield $v$.
4:     **if** there exists a neighbor $u$ of $v$ not in $S_1 \cup S_2$ **then**
5:         Choose $u$ such that edge $(u, v)$ has the smallest possible order and push $u$ to $S_1$.
6:     **else**
7:         Pop $v$ from $S_1$ and push $v$ to $S_2$.
8:     **end if**
9: **end while**

---

**Algorithm for IsTree.** We define Algorithm 1 as a depth-first search algorithm that can generate a sequence of vertices of length $O(n)$ that can be used to solve IsTree. We will use two stacks $S_1, S_2$ to store the vertices. $S_1$ will be used to store the vertices that are not yet visited, and $S_2$ will be used to store the vertices that are already visited. The algorithm will start with $S_1 = [v_1]$ and $S_2 = \emptyset$. At each step, the algorithm will pop the top of $S_1$ and push it to $S_2$. Then it will push all the neighbors of the popped vertex that are not in $S_1 \cup S_2$ to $S_1$. The algorithm will terminate when $S_1$ is empty. We will denote the yielded vertice sequence for $G$ as $\mathcal{A}(G)$. The following lemma shows the connection between the result of the algorithm and the IsTree problem.

**Lemma E.29.** *For any graph $G$, $\mathcal{A}(G)$ is a tree traversal of a spanning tree of the connected component of $G$ containing $v_1$. Hence $\mathrm{IsTree}(G)$ is True if and only if $G$ has $n-1$ edges and $\mathcal{A}(G)$ contains $2n - 1$ vertices.*

*Proof of Lemma E.29.* First, every vertex in the connected component of $G$ containing $v_1$ will be visited. This is because the algorithm will always push all the neighbors of the popped vertex that

are not in $S_1 \cup S_2$ to $S_1$. Hence, the algorithm will terminate when all the vertices in the connected component of $G$ containing $v_1$ are visited.

Second, every two consecutive vertices in the yielded sequence will be connected by an edge. This is because the algorithm will always push one of the neighbors of the popped vertex that is not in $S_1 \cup S_2$ to $S_1$. Hence, every two consecutive vertices in the yielded sequence will be connected by an edge. On the other hand, the combination of these edges will form a tree because the algorithm will never push a vertex that is already in $S_1 \cup S_2$ to $S_1$. Hence, the yielded sequence is a tree traversal of a spanning tree of the connected component of $G$ containing $v_1$. □

**Construction of Transformer.** We will now show that the yielded sequence of Algorithm 1 can be generated by a Transformer with constant depth and width, and $O(\log n)$ precision as a Chain of Thought. The Transformer will generate a valid yielded sequence but can terminate early if the graph is not a tree. We will now describe the Transformer in detail. We will assume the input token sequence $\mathcal{S}$ is as follows,

$$\mathcal{S} = \text{Tokenize}(G), v_1, \ldots v_r \tag{10}$$

for some $r \geq 0$ and $v_1 \ldots v_r$ is a valid yielded sequence. The length of $\text{Tokenize}(G)$ is $3n - 2$ with 3 tokens for each edges and 1 special token ``. We will further denote the input to the first layer $X$ as $\text{Emb}(\mathcal{S})$. We will similarly denote the input to layer $\ell$ as $X^{(\ell)}$. We will also denote the output of the last layer as $X^{out}$.

1. **Layer 1 and Layer 2 Attention.** The attention at Layer 1 will output zero and the FFN at Layer 1 and Attention at Layer 2 will implement a counting function (Definition E.8) to count the number of vertices $n$ appears in the previous token sequence and write $\text{ROUND}\left(\frac{1}{n}, p\right)$ in a new dimension $i_1$ as a result.

2. **Layer 2 FFN and Layer 3 Attention.** The FFN at Layer 2 and Attention at Layer 3 will implement a copying function (Definition E.7) copying the first dimension and the counting dimension $i_1$ of each token to its successor at two new dimensions $i_2$ and $i_3$. For each edge, this moves the type of the first vertice and the number of times the first vertice appears to $\sim$. For every vertice in the chain of thought, this moves the type of the previous vertice to them.

3. **Layer 3 FFN and Layer 4 Attention.** The FFN at Layer 3 and Attention at Layer 4 will implement another copying function, copying the dimensions $i_2$ and $i_3$ of each token to its successor at two new dimensions $i_4$ and $i_5$. Especially, for each edge, this moves the type of the first vertice and the number of times the first vertice appears to the position corresponding to the second vertices.

4. **Layer 4 FFN.** This FFN will process the information gathered from the previous layer and prepare for the next layer. It will make sure the following properties hold for $X^{(5)}$,

   - For every token, the position number, its square, and 1 will be kept in the last three dimensions.
   - For the first vertices in each edges, $\sim$ and `` The rest dimension will be zero.
   - For the second vertices of each edges $(a, b)$, there will be four dimensions $i_6, i_7, i_8, i_9$ with value $a, b$ and $n_{a,e}, n_{b,e}$, where $n_{a,e} = \text{ROUND}(\frac{1}{1 + \#a\text{appears up to current edge}}, 1)$.
   - For vertice $v_l$ in $v_1, \ldots, v_r$, there will be four dimensions $i_{10}, i_{11}, i_{12}, i_{13}$ with value $v_l, v_{l-1}$ and $v_l^2, v_{l-1}^2$ ($v_0 = 0$).

5. **Layer 5 Attention.** Combining with the previous Layer 4 FFN layer, we will implement two match functions with two attention heads matching $(i_{10}, i_{11})$ or $(i_{11}, i_{10})$ with $(i_6, i_7)$ at Layer 5 Attention, i.e. finding the edge in input for each step in the chain of thought, we will then copy $n_{v_l,(v_l,v_{l-1})}$ to dimensions $i_8$ and $i_9$.

6. **Layer 6.** We will use Layer 5 FFN and Layer 6 Attention to implement the match function that matches dimension $i_{10}$ of the current token to $i_{10}$ in the previous token. This will match $v_l$ to the first appearance of $v_l$ in the chain of thought and we will copy $i_{11}$ of the matched token to $i_{22}$. This dimension will be the first predecessor of $v_l$ in the chain of thought (0 for $v_1$). We will denote this predecessor of $v_l$ to be $f(v_l)$ as it is the father of $v_l$ in the tree. Now we will need to split into two cases depending on whether $v_{l-1}$ is $f(v_l)$. If $v_{l-1} = f(v_l)$ or $v_{l-1} = 0$ (for $v_1$), we will set dimension $i_8$ to be 1 and $i_9$ to be 0. Otherwise, we will keep dimension $i_8$ and $i_9$ as $n_{v_l,(v_l,v_{l-1})}$.

7. **Layer 7.** Now we will use Layer 6 FFN and Layer 7 Attention with two attention heads to implement two $\mathrm{MatchNext}$ functions[2] (Definition E.12) which use $i_8$ or $i_9$ as the counting index, and match $v_l$ at $i_{10}$ to $i_6$ or $i_7$ respectively. We will then copy dimensions $i_6$ to $i_9$ of the matched tokens to $i_{14}$ to $i_{21}$ (because there will be two of them).

The match next function will be able to retrieve the first edge containing $v_1$. For any $i \geq 2$, one of the matches next function will be able to retrieve the next edge containing $v_i$ after $(v_i, v_{i+1})$ if it exists. If it doesn't exist, the corresponding counting dimension will either be zero or no smaller than $n_{v_l,(v_l,v_{l-1})}$. We will use Layer 6 FFN to decide whether the next edge exists and set dimension $i_{14}$ of the output of Layer 6 to be the other edge in the next edge if it exists, or $0$ otherwise, and $i_{15}, i_{16}$ of the output of layer 6 to be the counting dimension of the next edge if it exists, or $0$ otherwise. For each edge in the original input, we will also set dimension $i_{15}, i_{16}$ to be the counting dimension of the edge.

8. **Layer 8 Attention** We will grab the next edge again, in the same manner as Layer 6, but this time using dimension $i_{15}$ and $i_{16}$. The necessity of this step is that the next edge containing $(v_{i-1}, v_i)$ in the original graph can be the same as the $(f(v_l), v_i)$ and in such case we need to check whether the next edge after this edge.

9. **Layer 8 FFN.** We now have, at each position corresponding to $v_l$, the first edge $(f(v_l), v_l)$ in the yielded sequence containing $v_l$ and the other vertex in the edge containing $v_l$ that hasn't been visited if it exists. If they don't exist, the corresponding dimension will be zero. This allows us to use Layer 8s FFN to decide the next vertex in the yielded sequence, which is exactly the first vertex different with $f(v_l)$ in the two edges if they exist, or $f(v_l)$ otherwise. We will use Layer 8 FFN to calculate the potential next vertex and put it in dimension $i_{23}$ and its square in $i_{24}$.

10. **Layer 9 Attention.** Combining with Layer 8 FFN, we will match $i_{23}$ of the current token to $i_{10}$ of the previous token to find the first appearance of the potential next vertex in the chain of thought. We will then copy dimension $d$ of the matched token to $i_{25}$. This value being $1$ will imply this vertex has never appeared in the chain of thought before and any other value will imply the vertex has appeared before.

11. **Layer 9 FFN.** We can now check several cases,

    - If the potential next vertex $v$ is either $f(v_r) \neq 0$ or never appears in the chain-of-thought sequence, then Layer 9 will output $n[-v, 1, -1, \ldots - 1]$, which will decodes to $v$.
    - If the potential next vertex $v$ is not $f(v_r)$ and appears in the chain-of-thought sequence, then Layer 9 will output $nw_6$, which will decodes to NO, because the chain of thought has already visited $v$ and hence the graph is not a tree.
    - If $v_r = 1$ and the potential next vertex $v$ is $f(v_r) = 0$, this means the chain of thought has finished. In this case, layer 9 FFN will check whether the position is $3n - 2 + 2n - 1 = 5n - 3$ and output $nw_5$ if it is, or output $nw_6$ otherwise, which will decode to YES and NO respectively.

This concludes the construction of the Transformer. $\qquad\square$

### E.7 PROOF OF THEOREM 4.8

The theorem is in the same vein as the recent work on the CoT for Transformer (Li et al., 2024), which shows the constant size and constant precision Transformer with a polynomial-size position embedding can simulate any polynomial size circuit. The major difference of our theorem is that (1) we consider a Transformer with fixed word and position embedding, hence allowing the parameter number to be logarithmic in the input size, and (2) we consider simulating RNNs, which is a special kind of circuit family and hence we can use more succinct representation utilizing the structural property attached to the recursive process.

We will now prove Theorem 4.8. We will first restate the theorem for convenience.

**Theorem 4.8.** *Given input length $n$, let $R$ is an RNN with word embedding $W^{(E)} \in \mathbb{R}_p^{(n+n_S) \times d}$, where $p = \Theta(\log n)$ is the precision, the constant $n_S$ is the number of special symbols in the*

---

[2]The constant $A$ here in Definition E.12is 1

*vocabulary, the constant $d$ is the embedding dimension. If each recurrent iteration can be computed by a circuit of size $C(n) \leq 2^{p/2}$, and if the RNN produces the final answer after running at most $n^A$ steps of CoT for some constant $A > 0$, then there exist Transformers with $O(\log n)$-bit precision, $O(C(n))$ parameters and word embedding $\begin{bmatrix} W^{(E)} & \mathbf{0}^{(n+n_S) \times d} \end{bmatrix}$ that can produce the same final answer after running $(C(n) + 1)n^A$ steps of CoT.*

*Proof.* The proof is inspired by Theorem E.30 from Li et al. (2024).

**Theorem E.30** (Theorem 3.3 of Li et al. (2024)). *For any $n$ and any circuit with size $T(n)$ and input size $n$, there exists a Transformer with constant depth and precision, $O(\log n)$ width, and a position embedding with size $O(T(n) \log n)$, such that for any input $\mathcal{S}$ of length $n$, the Transformer computes the output of the circuit on $\mathcal{S}$ using $T(n)$ steps.*

However, direct utilization of Theorem E.30 is not feasible because we are interested in (1) $O(\log n)$ precision Transformer, and (2) simulating the RNN for $n^A$ step, which would correspond to a circuit with $n^A C(n)$ in size. However, as the calculation is recurrent, we can encode the RNN circuit in $O(C(n))$ parameter instead.

To do so, we will unroll the circuit of each recurrent step of the RNN into $C(n)$ gates. We will then assign each gate a unique id in $[C(n)]$ and assume the circuit is calculated in the order of the gate id in the following manner.

1. Each gate has a type $t(i)$, which is either a constant gate outputting 1, an input gate, a hidden state gate, an AND gate, or an XOR gate.

2. Each gate $i$'s output depends on two values $l(i)$ and $r(i)$. If $t(i)$ is a constant gate, then $l(i)$ and $r(i)$ are assigned to be 0. When it is an input gate, $l(i)$ will be assigned to be the coordinate of the input embedding and $r(i)$ will be assigned to be the index of the bit of the value at $l(i)$ coordinate. When it is a hidden state gate, $l(i)$ will be assigned to be the coordinate of the hidden state embedding, and $r(i)$ will be assigned to be the index of the bit of the value at $l(i)$ coordinate. If it is an AND gate or an XOR gate, $l(i)$ and $r(i)$ will be assigned to be the id of the two gates that it depends on.

We will further assume without loss of generality that the hidden state gate is the first $p\Lambda$ gate. The output of the last $p\Lambda$ gate will be the next hidden state. We will also assume that the last $p(\Lambda + d)$ to $p\Lambda - 1$ gates are the output gates. We will now first describe the chain of thought that the Transformer will output and then construct the Transformer.

**Chain of thought**    Taking any input $\mathcal{S}$ with length $n$, the Transformer will output a sequence of 0 and 1 tokens. The first $n$ tokens will be the same as the input sequence. For each $a \geq 0$ and $b \in [C(n) + 1]$, the $n + a(C(n) + 1) + b$ token is

1. the output of gate $b$ when RNN circuit is calculating the output at $a$ position plus 1, if $b \leq C(n)$.

2. the $n + a + 1$ token in the RNN chain of thought, if $b = C(n) + 1$.

**Construction of the Transformer.**

1. **Layer 1.** The first attention layer will output zero and the first FFN layer will be of width $O(C(n))$, encoding all the gate information. The output of the first layer at position $n + a(C(n) + 1) + b$ will have the following coordinate:

   - The input $i$ will be encoded in the first dimensions.
   - $a, a^2, b, b^2$ will be encoded in four different dimensions.
   - The gate type $t(s(b))$ will be encoded in the next dimension, where $s(b) = (b + 1) \mod (C(n) + 1)$ If $b = C(n) - 1$, then the gate type will be encoded as 0.
   - The necessary dependence $l(s(b)), l^2(s(b))$ and $r(s(b)), r^2(s(b))$ will be encoded in the next two dimensions.

- A constant 1 will be encoded in the next dimension.

2. **Layer 2 Attention.** Together with the Layer 1 FFN, the Layer 2 Attention will implement two match functions (Definition E.9) to copy the output of gate $l(b+1)$ and $r(b+1)$ when RNN circuit is calculating the output at $a$ position. When the type of gate $b+1$ is not AND or XOR, the result will not be used in the following calculation.

3. **Layer 2 FFN** Layer 2 FFN will be of width $O(1)$. The output of the layer will be

   - When $b < C(n)$ and $t(s(b))$ is AND or XOR or constant, one dimension of the output will be the output of gate $b+1$ when RNN circuit is calculating the output at $a$ position.
   - When $b < C(n)$ and $t(s(b))$ is an input or hidden state gate or $b = C(n) + 1$, one dimension of the output will be the position in the current chain of thought where the input bit or hidden state bit is copied from and the other dimension will be the square of that position
   - When $b = C(n)$, the output remains the same as the input to Layer 2 FFN.

4. **Layer 3 Attention.** Layer 3 Attention will be of width $O(1)$. Together with Layer 2 FFN, Layer 3 Attention will implement $\mathrm{Match}$ heads (Definition E.9) to copy the output at the position where the input bit or hidden state bit is copied from. When the type of gate $b+1$ is not input or hidden state gate, the result will not be used in the following calculation.

5. **Layer 3 FFN** Layer 3 FFN will be of width $O(1)$. The output of the layer will be

   - When $b \neq C(n)$, one dimension of the output will be output of gate $s(b)$ when RNN circuit is calculating the output at $a + \mathbf{1}[b = C(n) + 1]$ position.
   - When $b = C(n)$, the output remains the same as the input to Layer 3 FFN.

6. **Layer 4** Layer 4 Attention will have $p - 1$ heads and each head will be of width $O(1)$. Head $h \in [p-1]$ will copy the first dimension of the output of Layer 3 FFN at position $n + a(P(n) + 1) + b - (p - h)$ and weight each of them by $2^{-h + (p-1)/2}$ and add them in one dimension. The Layer 4 FFN will calculate $r$ when the first dimension of the input is 1 and $-r$ otherwise. Hence, for each $a \geq 0$, the $n + a\left(C(n) + 1\right) - hp, h \in [\Lambda : \Lambda + d]$ token contains a dimension $i_1$ which is the $k - \Lambda$ dimension of the output of the RNN at position $a$.

7. **Layer 5** Layer 5 Attention will have $d + 1$ heads and each head will be of width $O(1)$. Head $h \in [d+1]$ will copy the dimension $i_1$ of the output of Layer 4 FFN at position $n + a(P(n) + 1) + b - (h + \Lambda)p$ to a disjoint dimension $i_{h+1}$. The Layer 5 FFN will then make sure the output of Layer 5 satisfies the following:

   - When $b \neq C(n)$, one dimension of the output will be $n$ times the output of gate $s(b)$ when RNN circuit is calculating the output at $a + \mathbf{1}[b = C(n) + 1]$ position plus 1, which will decode to the corresponding value.
   - When $b = C(n)$, the first $d$ dimension of the output will be the same as the output of the RNN at position $a$, and the rest dimension will be 0, which will decode to the same token as the chain of thought of the RNN at position $a + 1$.

This concludes the construction of the Transformer. □

## E.8 AR FUNCTION CALLS IS NOT ENOUGH FOR CLOSING THE REPRESENTATION GAP

**Proposition E.31.** *For any RNN family with $O(\log n)$ bit memory and $O(\log n)$ parameter with an oracle to receive results for the AR problem (Definition 4.3) for any queries, for large enough $n$, the RNN can't solve the index problem (Definition 4.2) with length $n$ in any CoT steps.*

*Proof.* Consider a special type of index problem where every token at the even position of the input sequence is a special token $\kappa$ and the rest of the tokens are uniformly random. Then the oracle for the AR problem can be simulated by the RNN by simply outputting the $\kappa$ when the query is not $\kappa$ and outputting the third token when the query is $\kappa$. However, following similar proof of Theorem 4.6, we can show that the RNN can't solve this special form of index problem with length $n$ in any CoT steps. □

### E.9 PROOF OF THEOREM E.32

We will first state the theorem for clarity.

**Theorem E.32.** *For task $T \in \{$Index, AR, c-gram retrieval, Counting$\}$, there exists a Linear RNN family with $O(\log n)$ bit memory and $O(\log n)$ parameter, that can solve $T$ with In-Context RAG in $O(1)$ CoT steps.*

*Proof.* When task $T$ is Index, given an input sequence that ends with a query token $k$, RNN will generate the following search query sequence:<StartSearch>, `^(?:\S\s*){`$k-1$`}(\S)`, <EndSearch>.

Then the regular expression will match the $k$-th token in the input sequence. The RNN needs to recite the format of the query, remember the index $k$ and calculate $k-1$ to generate the regular expression. As we have shown in Lemmas E.20 and E.23, RNN can recite a fixed sequence at fixed position and memorize the recent input, the above sequence can be generated by an RNN. The explicit construction of the RNN is omitted here.

When task $T$ is AR, given an input sequence that ends with a query token $k$, RNN will generate the following search query sequence: <StartSearch>, `\b`$k$`\b(\S+)\b`, <EndSearch>.

Then the regular expression will match the next token after the first occurrence of $k$ in the input sequence. The RNN needs to recite the format of the query and remember the query token $k$ to generate the regular expression. The explicit construction of the RNN is omitted here.

When task $T$ is c-gram retrieval, given an input sequence that ends with query tokens, RNN will generate the following search query sequence: <StartSearch>, `\b`$k_1 \ldots k_{c-1}$`\b(\S+)\b`, <EndSearch>.

Then the regular expression will match the next token after the first occurrence of $k_1, \ldots k_{c-1}$ in the input sequence. The RNN needs to recite the format of the query and remember the query tokens $k_1, \ldots k_{c-1}$ to generate the regular expression. The explicit construction of the RNN is omitted here.

When task $T$ is Counting, given an input sequence that ends with a query token $v$ and a query threshold $k$, RNN will generate the following search query sequence <StartSearch>, `(\b`$v$`\b){`$k+1$`}`, <EndSearch>.

Then the regular expression will match the $k$-th occurrence of $v$ in the input sequence. The RNN needs to recite the format of the query and remember the query token $v$ and the threshold $k$ to generate the regular expression. The RNN can then check whether the retrieval result is FAILED to determine if the count is less than $k$. The explicit construction of the RNN is omitted here. $\square$

### E.10 PROOF OF THEOREM E.33

In this section, we will prove Theorem E.33. We will first state the theorem for convenience.

**Theorem E.33.** *There exists a Linear RNN family with $O(\log n)$ bit memory and $O(\log n)$ parameter, that can solve IsTree of size $n$ with In-Context RAG in $O(n)$ CoT steps.*

*Proof of Theorem E.33.* We will first define the sequence that the retrieval-augmented RNN will generate and then construct an RNN that can generate such a sequence.

**Sequence Generation.** We will use a variant of Algorithm 1 to generate the sequence and we will use the concatenation of the tokenization of the sequence returned by Algorithm 2 as the sequence that the retrieval augmented RNN will generate.

**RNN Construction.** We can use similar proof in Theorem 5.2 by having the RNN memorize local sequences and determine the phase of Algorithm 2 it is in. The RNN will maintain the length of $S_2$ (Lemma E.21) and the top of $S_1$ in the state (Lemma E.20) and it is easy to check that the retrieval function will retrieve the correct result for each search query. The way to determine the next vertex in the stack is the same as in the proof of Lemma E.27. We will omit the simple but tedious detailed construction here. $\square$

---

**Algorithm 2** Depth-First Search Algorithm with Retrieving

---

**Require:** A graph $G = (V, E)$ with $n$ vertices and $E$ has an ordering $e_1, \ldots, e_m$.

1: Initialize two stacks of vertices $S_1, S_2$ with $S_1 = [v_1]$, $S_2 = \emptyset$, a list $L$ with $L = \emptyset$, and a vertex $v' = \text{FAILED}$.

2: **while** $S_1$ is not empty **do**

3:    Let $v$ be the top of $S_1$. Push $v$ to $L$.

4:    Generate the regular expression $r_2 = $ `\b(\S+)\b` $- $ `\b`$v$`\b` .

5:    Let $f(v)$ be the predecessor of $v$ in $S_1$ for the first time and FAILED when $v = v_1$.

6:    Push <StartSearch>, $r_2$, <EndSearch>, $f(v)$ to $L$.

7:    **if** $v' \neq f(v)$ **then**

8:       Generate the regular expression
$$r_1 \quad = \quad \left(\text{\b}v'\text{\b} \sim \text{\b}v\text{\b} \quad | \quad \text{\b}v\text{\b} \sim \text{\b}v'\text{\b}\right)\text{.*?}\left(\text{\b(\S+)\b} \sim \text{\b}v\text{\b} \quad | \right.$$
$$\left. \text{\b}v\text{\b} \sim \text{\b(\S+)\b} \right)$$

9:    **else**

10:       Generate the regular expression $r_1 = $ `\b(\S+)\b`$\sim$`\b`$v$`\b`$|$`\b`$v$`\b`$\sim$`\b(\S+)\b`

11:    **end if**

12:    Push <StartSearch>, $r_1$, <EndSearch> to $L$.

13:    **if** there exists a neighbor $u$ of $v$ such that $(u, v)$ has larger order than $(v, v')$ when $v' \neq f(v)$ or there exists a neighbor $u$ of $v$ such that $u \neq f(v)$ when $v' = f(v)$ **then**

14:       Choose $u$ such that edge $(u, v)$ has the smallest possible order and push $u$ to $L$. Let $v'' = u$.

15:    **else**

16:       Push FAILED to $L$. Let $v'' = \text{FAILED}$.

17:    **end if**

18:    **if** $v'' = f(v) \neq \text{FAILED}$ **then**

19:       Generate the regular expression
$$r_3 \quad = \quad \left(\text{\b}v''\text{\b} \sim \text{\b}v\text{\b} \quad | \quad \text{\b}v\text{\b} \sim \text{\b}v''\text{\b}\right)\text{.*?}\left(\text{\b(\S+)\b} \sim \text{\b}v\text{\b} \quad | \right.$$
$$\left. \text{\b}v\text{\b} \sim \text{\b(\S+)\b} \right)$$

20:       Push <StartSearch>, $r_3$, <EndSearch> to $L$.

21:       **if** there exists a neighbor $u$ of $v$ such that $(u, v)$ has larger order than $(v, v'')$ **then**

22:          Choose $u$ such that edge $(u, v)$ has the smallest possible order and push $u$ to $L$. Let $v'' = u$.

23:       **else**

24:          Push FAILED to $L$. Let $v'' = \text{FAILED}$.

25:       **end if**

26:    **end if**

27:    **if** $v'' = \text{FAILED}$ **then**

28:       Pop $v$ from $S_1$. Push $v$ to $S_2$. Let $v' = v$.

29:    **else**

30:       Generate the regular expression $r_4 = $ `\b(\S+)\b` $- $ `\b`$v''$`\b`

31:       Push <StartSearch>, $r_4$, <EndSearch>, $0$ to $L$.

32:       **if** $v''$ is not in $S_1$ **then**

33:          Push FAILED, $v, -, v''$ to $L$.

34:          Push $v''$ to $S_1$. Let $v' = v$.

35:       **else**

36:          Let $f(v'')$ be the predecessor of $v$ in $S_1$ for the first time and FAILED when $v'' = v_1$.

37:          Push $f(v''), \text{NO}$ to $L$.

38:          **return** $L$.

39:       **end if**

40:    **end if**

41: **end while**

42: **if** $S_2$ has $n$ vertices **then**

43:    Push YES to $L$.

44:    **return** $L$.

45: **else**

46:    Push NO to $L$.

47:    **return** $L$.

48: **end if**

---

### E.11 PROOF OF THEOREM 5.2

In this section, we will prove Theorem 5.2. We will first restate the theorem for convenience.

**Theorem 5.2.** *Given $A, B > 0$, for all polynomial-time Turing machines $T \in \mathrm{TIME}(n^A)$ with $B$ states and vocabulary size $B$, there exist Linear RNNs with $B$ special symbols, $O(A \log n)$-bit precision and memory, and $O(AB^2)$ parameters that can output the result of $T$ by running $O(n^A)$ steps of CoT with In-Context RAG.*

*Proof of Theorem 5.2.* We will denote the state of $T$ as $1, \ldots, B$ (we will use 1 as the initial state) and the vocabulary of $T$ as $1, \ldots, B$. We will assume $T$ operates on an infinite tape TAPE, which is a sequence of cells indexed by $\mathbb{Z}$. We will also assume that the tape is initialized with all cells being 0 except for the $n$ cell starting at 1. The Turing machine also has a pointer $p$ that points to a cell in the tape. The pointer is initialized to 1. At each time step, the Turing machine reads the cell pointed by POINTER and updates the cell pointed by POINTER and the pointer $p$ according to the transition function $\delta : [B+1] \times [B] \rightarrow [B] \times [B] \times \{-1, 1\}$, which takes the current state and the current cell value (could be empty, which corresponds to $B + 1$) as input and outputs the new state, the new cell value and the direction to move the pointer. The Turing machine halts when the state is $B$. Because $T \in \mathrm{TIME}(n^A)$, the Turing machine will always halt in $n^A$ steps. We will use $\mathrm{TAPE}[t, i]$ as the value on the $i$-th cell on TAPE before the $t$ timestep. We will use $\mathrm{POINTER}[t]$ as the value of the pointer before the $t$ timestep and $\mathrm{State}[t]$ as the state of the Turing machine before the $t$ timestep. We will further use $\mathrm{Direction}[t]$ as the direction of the pointer movement before the $t$ timestep.

We will first define the sequence that the retrieval-augmented RNN will generate and then construct an RNN that can generate such a sequence.

**Sequence generation.** The input token sequence $\mathcal{S}_{\mathrm{in}}$ will be as followed,

$$\mathcal{S}_{\mathrm{in}} = \texttt{}, \mathrm{TAPE}[1, 1], \mathrm{TAPE}[1, 2], \ldots, \mathrm{TAPE}[1, n]$$

Here all the symbols on the tape are represented by one special symbol in the vocabulary. Given this input token sequence, the retrieval augmented RNN will generate the following output token sequence,

$$\mathcal{S} = \mathcal{S}_{\mathrm{in}}, \texttt{<StartSearch>}, \texttt{\textasciicircum(?:\textbackslash S\textbackslash s*).\{1\}(\textbackslash S)}, \texttt{<EndSearch>},$$
$$\quad \textcolor{red}{\mathrm{TAPE}[1, 1]}$$
$$\quad 1, \mathrm{TAPE}[1, 1], 1,$$
$$\quad \ldots$$
$$\quad \texttt{<StartSearch>}, \texttt{\textasciicircum(?:\textbackslash S\textbackslash s*).\{n\}(\textbackslash S)}, \texttt{<EndSearch>},$$
$$\quad \textcolor{red}{\mathrm{TAPE}[1, n]}$$
$$\quad n, \mathrm{TAPE}[1, n], n,$$
$$\quad \texttt{<StartSearch>}, \texttt{((}\textcolor{blue}{\mathrm{POINTER}[1]}\texttt{ (.) }\textcolor{blue}{\mathrm{POINTER}[1]}\texttt{ .*?\$))}, \texttt{<EndSearch>},$$
$$\quad \textcolor{red}{\mathrm{SearchResult}(1),}$$
$$\quad \mathrm{POINTER}[1], \mathrm{TAPE}[2, \mathrm{POINTER}[1]], \mathrm{POINTER}[1]$$
$$\quad \mathrm{State}[2], \mathrm{Direction}[2],$$
$$\quad \ldots$$
$$\quad \texttt{<StartSearch>}, \texttt{(}\textcolor{blue}{\mathrm{POINTER}[t]}\texttt{ (.) }\textcolor{blue}{\mathrm{POINTER}[t]}\texttt{ .*?\$)}, \texttt{<EndSearch>},$$
$$\quad \textcolor{red}{\mathrm{SearchResult}(t)}$$
$$\quad \mathrm{POINTER}[t], \mathrm{TAPE}[t + 1, \mathrm{POINTER}[t]], \mathrm{POINTER}[t],$$
$$\quad \mathrm{State}[t + 1], \mathrm{Direction}[t + 1],$$

Here $\mathrm{SearchResult}(t)$ is defined as

$$\mathrm{SearchResult}(t) = \begin{cases} \mathrm{FAILED}; & \text{if } \mathrm{POINTER}[t] \text{ is empty cell before } t \\ \mathrm{TAPE}[t, \mathrm{POINTER}[t]]; & \text{otherwise} \end{cases}$$

The output token sequence simulates the Turing machine $T$ on the tape TAPE due to the following lemma.

**Lemma E.34.** *Given any $t \in [n^A]$ and $i \in [n^A]$, the last string in $\mathcal{S}$ that contains $i, i$ as a substring is $i, \mathrm{TAPE}[t, i], i$ if $\mathrm{TAPE}[t, i]$ is not empty and is the empty string otherwise.*

*Proof.* The proof is by induction, for $t = 1$, the result holds. For any $t \geq 2$, we only need to notice that $\mathrm{POINTER}[t-1]$ is the only cell that can be updated at time $t-1$. $\qquad\square$

**Construction of RNN**  Given the input, the RNN can first iterate over $1$ to $n$ and generate the first $n$ search queries and results by maintaining a counter in its state and memorizing the most recent search result (Lemma E.20). Then it is easy to see that the retrieval oracle will generate the correct $\mathrm{SearchResult}(t)$ given the input $\mathcal{S}$. Therefore, we will only need to construct an RNN that can generate the rest part of $\mathcal{S}$.

We will assume the RNN maintains the state and pointer of the Turing machine in its state and show that they can be updated.

Based on Lemma E.20, the RNN can maintain constant recent token types in its state, we will assume the RNN memorize the last tokens up to the most recent <StartSearch> and also calculate the position relative to the most recent <StartSearch>. By a lookup table in the FFN Lemma E.4, the RNN can output the fixed format of the search query. Similarly, RNN can output the $\mathrm{POINTER}[t]$. To generate the update $\mathrm{TAPE}[t+1, \mathrm{POINTER}[t]], \mathrm{State}[t], \mathrm{Direction}[t]$, the RNN can use a FFN with $O(B^2)$ width to memorize the transition function of the Turing machine and output the update. Then, the RNN can use the memorized recent input to update the state and the pointer of the Turing machine at the next <StartSearch>. The proof is then complete.

$\qquad\square$

### E.12   PROOF OF THEOREM E.35

**Theorem E.35.** *For task $T \in \{$Index, AR, c-gram retrieval, Counting$\}$, there exists a hybrid Linear RNN (Definitions B.7 and B.8) family with $O(\log n)$ bit memory and $O(\log n)$ parameter, that can solve $T$ without CoT.*

*Proof.* The proof here is essentially the same as the construction of the Transformer in Theorem 4.6. We would use the same Transformer layer to solve $T$. The only difference is that we would use the output of the RNN, instead of FFN, as the input of the Transformer layer. Also for Counting, instead of using a COPY function, we write the query token in the state of the RNN (Lemma E.20). $\qquad\square$

### E.13   PROOF OF THEOREM E.36

**Theorem E.36.** *There exists a hybrid Linear RNN with $O(\log n)$ bit memory and $O(\log n)$ parameter, that can solve IsTree of size $n$ with a chain of thought of length $O(n \log n)$.*

*Proof.* The proof is similar to the proof of Theorem E.33. However, instead of using regular expressions to retrieve the next neighbor and parent, we will need to use the Transformer layer. The Transformer layer can retrieve the parent through an attention head implementing the match closest head (Lemma E.16) if the RNN part maintains the predecessor of each node in the chain of thought.

Retrieving the next neighbor is more complicated and we will use $O(\log n)$ steps of the chain of thought to do that. Given an edge $(v, v')$, we will first use one match head to retrieve the position $p$ of $(v, v')$ in the input sequence and write it to the chain of thought. Then we will use two $\mathrm{MatchClose}$ heads to retrieve the edge that contains $v$ and is closest to $p + 2^i$ for $i = 0, 1, \ldots, \log_2 n$ iteratively until the heads return an edge that is not $(v, v')$ or $i$ reaches $\log_2 n$. Here $2^i$ can be computed through doubling one of the dimensions in the state of the RNN and reset that dimension to $1$ after termination. We will then compare the retrieved edge with the father of $v$ to check if it is the same. If it is the same, we will search the next neighbor of $v$ after the parent of $v$ in the same way. The other part of the proof is similar to the proof of Theorem E.33. $\qquad\square$

## E.14 PROOF OF THEOREM 5.4

**Theorem 5.4.** *Given $A, B > 0$, for all polynomial-time Turing machines $T \in \text{TIME}(n^A)$ with $B$ states and vocabulary size $B$, there exist a constant-size hybrid Linear RNNs with $B$ special symbols, $O(A \log n)$-bit precision and memory, and $O(AB^2)$ parameters that can output the result of $T$ by running $O(n^A)$ steps of CoT.*

*Proof.* **Sequence Generation.** Under the same formulation of proof of the Theorem 5.2. The hybrid RNN will output the following sequence.

$$\mathcal{S} = \mathcal{S}_{\text{in}}, \text{POINTER}[1], \text{TAPE}[2, \text{POINTER}[1]], \text{POINTER}[1]$$
$$\text{State}[2], \text{Direction}[2],$$
$$\dots$$
$$\text{POINTER}[t], \text{TAPE}[t+1, \text{POINTER}[t]], \text{POINTER}[t],$$
$$\text{State}[t+1], \text{Direction}[t+1],$$

Note that Lemma E.34 still holds. We only need to prove that the hybrid architecture can generate the above sequence.

**Hybrid Construction.** The way RNN maintains the pointers and the states is the same as the proof of Theorem 5.2. Given each pointer value, we can retrieve the last value of the cell at the pointer through the one layer of attention by implementing a match closest head (Lemma E.16). □

