# OpenReview forum: "RNNs are not Transformers (Yet):  The Key Bottleneck on In-Context Retrieval"
_ICLR.cc/2025/Conference — ICLR 2025 Poster_

### Official Review · Reviewer_aZ5D · 2024-11-02

**Soundness:** 3
**Presentation:** 2
**Contribution:** 3
**Rating:** 5
**Confidence:** 3

**Summary:**

This paper investigates the gap in representation power between Transformers and Recurrent Neural Networks (RNNs) and examines whether RNNs, with techniques like Chain-of-Thought (CoT) prompting, can achieve comparable performance. Through theoretical analysis, the authors demonstrate that RNNs lack the expressiveness needed to handle in-context retrieval tasks, such as associative recall and graph recognition, that Transformers handle with ease. They further support these findings with empirical evidence from both synthetic tasks and natural language experiments, showing that while CoT prompting improves RNNs, it alone is insufficient to bridge the gap.
The authors propose that enhancing RNNs with Retrieval-Augmented Generation (RAG) or adding a Transformer layer can significantly close this performance gap, bringing RNNs closer to Transformer-level capabilities. However, they also acknowledge that practical challenges, such as computational costs and scalability, may arise when implementing these improvements, suggesting a trade-off between efficiency and retrieval effectiveness.

**Strengths:**

A key strength of this paper is its use of theoretical lower bounds to rigorously show that RNNs lack the expressiveness needed for certain in-context retrieval tasks, unlike Transformers. This analytical approach confirms that RNN limitations are inherent, not merely practical, underscoring the need for hybrid architectures or additional layers to bridge the performance gap. Additionally, the paper validates these findings through empirical experiments across various tasks (such as associative recall and graph recognition), demonstrating that the theoretical limitations are evident in real applications as well.
The authors also provide practical techniques—like Retrieval-Augmented Generation (RAG) and hybrid architectures involving Transformer layers—that make their findings actionable and relevant for architecture design in large language models (LLMs). By identifying specific ways to improve RNNs, the paper offers a path toward resource-efficient alternatives to Transformers, valuable for applications in memory-constrained or mobile environments. Moreover, their analysis sheds light on the mechanisms behind in-context learning, clarifying what capabilities make Transformers excel and highlighting specific limitations in RNNs. This deeper understanding could inform future research on in-context retrieval and neural network architecture design.

**Weaknesses:**

Readability and Structure: The paper is hard to follow due to poor language flow, making the argument difficult to track.
Redundant content, especially on Chain of Thought (CoT) prompting, appears in both the Related Work and Preliminaries sections. Consolidating these would improve readability. Some definitions are introduced but not utilized in the main text, adding unnecessary complexity.

Depth of Discussion and Empirical Validation: The paper discusses two main ideas: first, that Chain of Thought (CoT) prompting can improve RNNs but is insufficient to fully bridge the representation gap with Transformers; and second, that enhancing the in-context retrieval capability of RNNs can potentially close this gap. However, each of these points would benefit from deeper discussion and more practical experiments. The empirical validation is insufficient; model comparisons are not “apple-to-apple, limiting the clarity and robustness of results.

**Questions:**

None

---

> ### Author Response · Authors · 2024-11-23
>
> We thank the reviewer for stating our findings as actionable and relevant. We would like to address some of the concerns here.
>
> **W1: Need to improve readability.**
>
> We appreciate the reviewer's feedback on the readability of the paper and would like to address the concerns as follows.
>
> **W1.1: Poor language flow.**
>
> We have carefully proofread the paper to ensure clarity, but the comment on "poor language flow" lacks specificity for us to act on. Could the reviewer provide concrete sections or paragraphs where the language flow is problematic?
>
> **W1.2: Redundant content on CoT**
>
> The two mentions of CoT in our paper serve different purposes.  The first mention in the related works reviews prior works, and the second mention in the preliminaries introduces a formal definition of CoT.  We agree with the reviewer that it is not necessary to cite previous works on CoT prompting in the second mention and have removed the corresponding sentence.
>
> **W1.3: Unused definitions**
>
> We checked the text carefully and could not identify any definitions introduced but left unused in the main text. Could the reviewer clarify which definitions they believe were unnecessary?
>
> We would appreciate the opportunity to further discuss with the reviewer to refine the paper based on clarified and more actionable suggestions.
>
> **W2: The depth of discussion and empirical validation is not sufficient and is not ‘apple-to-apple’.**
>
> We disagree with the reviewer that the empirical validation is not sufficient to support our arguments and is not ‘apple-to-apple’.
>
> 1. On the synthetic side, we controlled the parameter size of each architecture rigorously on the IsTree task. We observe that Transformers and hybrid architectures outperform RNNs significantly on this task when the number of vertices increases. We also observed that chain-of-thought and retrieval augmented generation improved all the architectures we tested, further enhancing our theory.
> 1. In the real-world experiments on HotpotQA, due to the limited compute resources, we could not afford to pretrain models from scratch and had to use open-source pretrained models. However, to maximally guarantee we are close to making an *apple-to-apple* comparison, we tested on RNNs and hybrid models distilled from the pretrained Phi-1.5 models. In this way, we can guarantee that the models are of similar parameter sizes and go through approximately the same training dataset. The observed performance is highly consistent with our theoretical analysis.
>
> Combining the experiments in both settings, we believe the experiments conducted so far strongly supported our theoretical results.

---

### Official Review · Reviewer_4LG8 · 2024-11-03

**Soundness:** 3
**Presentation:** 3
**Contribution:** 3
**Rating:** 8
**Confidence:** 2

**Summary:**

This paper compares the performance differences between RNNs and Transformers when enhanced with chain of thought (CoT) reasoning. The term RNN is used in a broad sense to refer to the class of architectures that are linear time and constant memory with respect to sequence length. This paper demonstrates, both theoretically and empirically, that while CoT enhances the performance of RNNs, it is not sufficient to close the gap between RNNs and Transformers. To close this gap, the authors demonstrate that retrieval-augmented generation or adding a single transformer layer is sufficient.

**Strengths:**

1. This is a well-designed study with a focused question, along with theoretical and empirical support for its conclusions.
2. This paper’s theoretical analysis is a strong contribution, given the numerous prior works that empirically find hybridization and RAG help close the performance gap between transformers and RNNs.

**Weaknesses:**

1. Perhaps this is a misunderstanding, but line 297 seems to contradict Arora et al. (2023), who observe that while transformers outperform RNNs on associative recall, certain RNNs can perform associative recall. Claiming that “constant-size RNNs cannot solve any of the four tasks” seems too strong, given that Theorem 4.6 is only true for “large enough $n$.” This paper would benefit from the addition of a paragraph reconciling this claim with the empirical results of prior work.
2. The use of IsTree as a synthetic task could be better motivated. As currently written, it seems that this task was chosen simply because it does not obviously require retrieval to solve which is not unique to IsTree. It would be helpful to expand on the authors' rationale.

**Questions:**

1. Why were the word embedding and decoding heads frozen in the experiments for Figures 2 and 4?
2. How do the author's define a "RNN with o(n)-bit memory"?

---

> ### Author Response · Authors · 2024-11-23
>
> **W1: Does the current result contradict Arora et al. (2023) which states that empirically some RNNs can perform associative recall?**
>
> There is no contradiction because we are proving the performance when the context length is long enough and Arora et al. are only testing on a maximal context length of 512\. Some related empirical studies can justify our results: in \[1\], both pretrained Mamba and Mamba-2 7B models are shown to have significantly worse Phonebook-retrieval capabilities on 1K context length than Transformers with the same size and trained on the same data. We have added a paragraph discussing how our results connect with prior empirical observations.
>
> \[1\] An Empirical Study of Mamba-based Language Models
>
> **W2: What is the reason why we pick the task IsTree?**
>
> Here, we briefly argue why we prefer the IsTree task over other synthetic tasks.
>
> 1. As the reviewer mentioned, IsTree is a simple and well-known algorithmic problem that is seemingly irrelevant to in-context retrieval. However, as a classical problem studied in the literature of streaming algorithms, IsTree has been proven to require $O(n)$-bit memory for streaming algorithms, where $n$ is the number of nodes.
> 2. IsTree serves as a minimal abstraction of logical reasoning suggested in line 987\.  In a logical reasoning setting, if we view theorems as vertices and equivalences as edges, when given n theorems and n−1 relations, checking the equivalence of all the theorems reduces to verifying whether the graph forms a tree. This connection highlights IsTree's relevance as a foundational task for studying logical reasoning.
>
> **Q1: Why were the word embedding and decoding heads frozen in the experiments for Figures 2 and 4?**
>
> In line with our theoretical setup, we consider a large vocabulary that contains all the numbers in $1$ and $n$ (where n is the maximum input length). Asymptotically, this will lead to a model size linear in the input length for large $n$. To ensure that the model does not have too many parameters, we fix the word embedding and decoding heads as a function that deterministically maps numbers to vectors. We believe our experiment results still hold even if we use a more standard tokenizer that decodes numbers into several digits.
>
> **Q2: How do the authors define a "RNN with o(n)-bit memory"?**
>
> We say that an RNN has $M$-bit memory if we need $M$ bits to represent the internal state of the RNN. In other words, if there are $S$ floating-point numbers in the state and each number is represented by $f$ bits, then the RNN has $M \= S \\cdot f$ memory.

---

> > ### Comment · Reviewer_4LG8 · 2024-12-02
> >
> > > There is no contradiction because we are proving the performance when the context length is long enough and Arora et al. are only testing on a maximal context length of 512. Some related empirical studies can justify our results: in [1], both pretrained Mamba and Mamba-2 7B models are shown to have significantly worse Phonebook-retrieval capabilities on 1K context length than Transformers with the same size and trained on the same data. We have added a paragraph discussing how our results connect with prior empirical observations.
> >
> > The original issue I pointed out is in line 296 of the latest revision, not in Theorem 4.6: "constant-size RNNs cannot solve any of the four tasks." This claim is made immediately after defining index, associative recall, etc., but your definition does not specify for which values of $n$. Therefore, I still contend that it is easy for readers to assume that you are claiming that RNNs cannot perform associative recall at all, though the new paragraph does help add some clarity. You could also just qualify line 296 instead of updating your definitions.

---

> > > ### Author Response · Authors · 2024-12-02
> > >
> > > We agree with the modification and will quantify line 296 in future versions. Thank you for the valuable suggestion!

---

### Official Review · Reviewer_8brv · 2024-11-04

**Soundness:** 3
**Presentation:** 2
**Contribution:** 3
**Rating:** 6
**Confidence:** 3

**Summary:**

This paper investigates the representational limitations of Recurrent Neural Networks (RNNs) compared to Transformers in handling sequence modeling tasks. It particularly examines RNNs’ in-context retrieval capabilities, which are essential for tasks requiring memory and context comprehension. Through theoretical analysis and empirical validation, the authors identify the inherent weaknesses of RNNs in retrieving information across long sequences and propose potential methods to bridge the performance gap between RNNs and Transformers.

**Strengths:**

Originality:
The paper tackles a unique question—whether RNNs can be enhanced to achieve Transformer-like in-context retrieval. By applying communication complexity and information-theoretic methods, it provides a fresh perspective on RNN limitations. Proposing hybrid architectures and Retrieval-Augmented Generation (RAG) is an innovative approach to bridging this gap.

Quality:
The paper is theoretically rigorous, with strong proofs demonstrating RNN limitations. The authors conduct well-designed empirical experiments that validate the theoretical claims on both synthetic and real-world tasks, making the results robust and credible.

Clarity:
The structure is logical, with clear definitions and effective visuals that aid understanding. Complex proofs could benefit from simplified explanations, but overall, the writing is accessible and well-organized.

Significance:
This work addresses a critical issue in model efficiency, with practical implications for memory-constrained applications. It lays a solid theoretical foundation for future research on enhancing RNNs, potentially impacting both low-resource applications and the development of hybrid model architectures.

**Weaknesses:**

1.This paper spends a large section explaining the well-known concepts in AI community like Language Models and CoT while ignoring the explanation of communication complexity and other less-known concepts that are also important in the proof. It would be better to shift the focus of concept explanation.

2. In some cases, with enough training, RNN may have the ability to better organize information based on the original O(n) bit. For example, based on the case of in-context retrieval, it may be possible to finetune the RNN to enable it to achieve the information retrieval process with the deepening of layers. e.g. the hidden states in shallow layers store the raw information and the deeper layers add organized information (like the regular expression for retrieval ) into the hidden state.  Please disprove this possibility.

**Questions:**

Why in-context retrieval can not be deemed as a part of COT?
Since I tried RWKV6 -7B and it can retrieve in context words without the help of regular express function but with the help of CoT.
I tried api on https://huggingface.co/spaces/BlinkDL/RWKV-Gradio-2
My input is:

Assistant: ”How can we craft an engaging story featuring vampires on Mars? Let's think step by step and provide an expert response. “ What's the word between 'featuring' and 'on'. Think step by step.

The result is unstable but primarily consist of 'vampire' and 'mars‘， which means RWKV has somewhat retrieval ability.

---

> ### Author Response · Authors · 2024-11-23
>
> We thank the reviewer for praising the originality and clarity of this paper and would like to address the concerns here.
>
> W1: The paper should shift to explain communication complexity and other less-known concepts.
>
> We would like to thank the reviewer for the valuable suggestion. We have updated our related work section to explain the relationship between communication complexity and streaming algorithms which are important for the proof. We will also add a more detailed explanation in future versions.
>
> W2: RNNs can learn to organize and compress the context to break the theoretical lower bound.
>
> While RNNs can and will learn an efficient compression for some prefixes, our paper argues that there exist important atomic tasks like in-context retrieval that require a state size that grows linearly with the context length. This lower bound will hold regardless of the depths of RNNs and couldn’t be solved through finetuning. At a high level, this is because the RNNs could only write the information of the prefix into the constant-size state but information-theoretically in-context retrieval requires a large state size.
>
> Q1: Why in-context retrieval can’t be a part of CoT as models like RWKV can retrieve from context?
>
> We do not claim that RNNs couldn’t retrieve information from context of any length. Instead, we show that the context length that RNNs can retrieve the context perfectly from is linearly scaling with its state size. Therefore, it is not a surprise that empirically RNNs can retrieve from short context as provided in the question because the state size of the model may be large enough to compress the short context into the state already. Our theory further shows that this limitation is not amenable by any type of chain-of-thought, including asking the model to explicitly retrieve from the context.

---

> > ### Comment · Reviewer_8brv · 2024-12-03
> >
> > Thank you for your response. Most of my concerns have been addressed.

---

### Official Review · Reviewer_bMYq · 2024-11-04

**Soundness:** 2
**Presentation:** 3
**Contribution:** 1
**Rating:** 3
**Confidence:** 4

**Summary:**

This paper investigates the representational capabilities of RNNs and Transformers, both with and without Chain-of-Thought (CoT). Assuming a limited RNN state size of O(log n), the authors show that RNNs augmented with CoT have greater representational power than standard RNNs, but are still less powerful than Transformers with CoT. To bridge this gap, the authors propose adding in-context retrieval mechanisms to RNNs, showing that this extension enables RNNs with CoT to solve polynomial-time problems, effectively matching the capabilities of Transformers with CoT.


Informally, the relationships can be summarized as:
- Standard RNN < RNN + CoT < Transformer + CoT
- RNN + In-Context Retrieval + CoT ~= Transformer + CoT

**Strengths:**

- The paper is well-structured and clearly written, main ideas and results are easy to follow.

- The paper constructs building blocks for Transformers that can be used to solve various tasks, including indexing, associative recall, c-gram retrieval, and counting. It provides detailed analyses of how these problems can be addressed using different architectures and shows their limitations.

- The authors support their theoretical results with empirical evaluations on tasks such as IsTree and HotPotQA, thus strengthening their claims.

- Show that in-context retrieval, similar to attention mechanism, is sufficient to close the gap between RNNs and Transformers with CoT, and suggest how to build more expressive RNN architectures.

**Weaknesses:**

1. The authors compare RNNs and Transformers under different assumptions regarding their state sizes. Specifically, they limit RNNs to a limited hidden state size of O(log n), while allowing Transformers to have an effective state size that grows linearly with the input length due to the accumulation of key-value pairs in self-attention. A decoder-only Transformer can be viewed as an RNN with a recurrent state that increases linearly -- adding each processed input to the key-value cache:  in transformer h_t = Transformer(x_t, [h_1, …, h_{t-1}]), in RNN h_{t}  = RNN(x_t, h_{t-1}); In other words, the results obtained are equivalent to comparing RNNs with limited state sizes to RNNs with linearly growing state sizes. There is nothing inherently specific to Transformers in this comparison. The key differentiator in this comparison is the state size rather than any unique property of Transformers. Therefore, concluding that models with larger state sizes have more expressive power than those with limited state sizes is unsurprising.

2. Violation of initial assumptions with in-context retrieval. By introducing in-context retrieval to RNNs, the model effectively needs to store the entire input sequence, resulting in O(n) state size requirements - similar to Transformers. This breaks the initial assumption of a limited-size RNN state O(log n) used to state RNNs limitations. As a result, the proposed solution does not address the original problem under the same constraints.

3. The proposed solution of adding in-context retrieval to RNNs essentially increases the state size linearly, effectively transforming the RNN into an RNN with linearly growing state size, which resembles a decoder-only Transformer. This leads to the somewhat tautological conclusion that to close the gap between RNNs and Transformers, one should make a Transformer-like model out of an RNN.

4. Overlap with existing concepts is not discussed. The relationship between RNNs with CoT and Adaptive Computation Time (ACT) mechanisms is not fully explored. Since CoT can be seen as a form of ACT, and RNNs with ACT have been previously studied, the novelty of the findings may be limited. A deeper exploration of how this work builds upon or differs from existing research on RNNs with ACT could strengthen the contribution.

**Questions:**

Please address points from weaknesses as questions if authors do not agree.

---

> ### Author Response · Authors · 2024-11-23
>
> We thank the reviewer for acknowledging this paper as being well-written and would like to address the reviewer’s concern here.
>
> **W1.1: The authors compare RNNs and Transformers under different assumptions regarding their state sizes.**
>
> The reason that we compare RNNs and Transformers with different state sizes is that RNNs indeed have smaller state sizes than Transformers with the same number of parameters.
>
> 1. Having a smaller state size is a key feature of RNN, and it has motivated the recent empirical trend of developing modern RNNs (such as Mamba) to avoid the high memory cost of Transformers on long context inputs.
>    1. For example, in \[1\], it is argued that *“RNNs require less memory, particularly for handling long sequences.”*
>    2. Our work reveals a fundamental limitation of this approach, showing that such models with high memory efficiency are incapable of doing in-context retrieval.
>    3. To overcome this limitation, the state size of RNN needs to scale linearly with respect to the context length for in-context retrieval.
>    4. Given Transformer’s state size scales linearly with the context length, it already reaches the optimal asymptotic rate when the context length grows so it is impossible to improve the memory efficiency beyond a constant factor.
> 2. We compare RNNs and Transformers with the same order of number of **parameters** because it is typical in current empirical studies to test modern RNN architectures on in-context retrieval tasks with Transformers with the same amount of parameters \[1,2,3,4\]**.** More specifically, we assume the parameter size of both architectures to be constant and the precision is of order $\\log n$ when the context length is $n$. This is a standard assumption in previous works considering the representation power of Transformers \[5, 6\].
>
> \[1\] RWKV: Reinventing RNNs for the Transformer Era
> \[2\] Mamba: Linear-Time Sequence Modeling with Selective State Spaces
> \[3\] Transformers are SSMs: Generalized Models and Efficient Algorithms Through Structured State Space Duality
> \[4\] Parallelizing Linear Transformers with the Delta Rule over Sequence Length
> \[5\] Towards Revealing the Mystery behind Chain of Thought: A Theoretical Perspective
> \[6\] Self-Attention Networks Can Process Bounded Hierarchical Languages
>
> **W1.2: Transformers can be viewed as RNNs with linear state size, and hence showing Transformers outperforming RNNs with state size $O(\\log n)$ is not surprising.**
>
> 1. Models with larger state sizes do not automatically imply a stronger representation power. To prove our results, some key details in the Transformer architecture are crucial. For example, if the softmax mechanism is removed, the linear attention mechanism will not be able to perform in-context retrieval with context length $n$. This is because such Transformers are not effectively using their large state sizes. In fact, one can rewrite them as RNNs with $O(\log n)$ state sizes.
> 2. Can we rewrite Transformers with softmax attention as small RNNs? This is unclear a priori, but our work is able to show that Transformers with softmax attention are more expressive than RNNs, especially on in-context retrieval tasks. So there is something *inherently specific to Transformers (with softmax attention) in this comparison.*
> 3. We not only show that RNNs are worse than Transformers on the simple in-context retrieval tasks but also show that even if we use CoT, RNNs are still worse than Transformers (without CoT).

---

> ### Author Response · Authors · 2024-11-23
>
> **W2&3: Augmenting the model to retrieve from context violates the initial assumption as these augmentations enlarge the state size. This makes the argument tautological.**
> We respectfully disagree that these conclusions are tautological for the following reasons.
>
> 1. First, we would like to clarify our main results on enhancing the in-context retrieval capability of RNNs, as we suspect that the reviewers may have misunderstood our results. We proposed two approaches: In-context RAG and adding one Transformer layer.
>    1. For the first approach, we are still considering RNNs with O(log n) state size, in contrast to the linear state size mentioned by the reviewer. However, the reviewer is right because the whole inference algorithm needs O(n) memory to implement the In-context RAG externally.
>    2. For the second approach, it is obvious that adding 1000 Transformer layers to an RNN can greatly enhance its representation power, so as mentioned in the introduction (Line 137), we are interested in the minimal possible change to the architecture that can close the representation gap. Our final result shows that adding 1 layer can just work.
> 2. Both the above approaches use O(n) memory, but we argue that with o(n) memory, no approach can perform in-context retrieval. The reason is exactly what we have shown in the first part of our paper (Section 4).
> 3. Our results are not tautological because not all approaches with O(n) memory can perform in-context retrieval.
>    1. A concrete counterexample is RNN augmented with an associative recall oracle. As stated in line 382 and Appendix E.8 of our paper, this cannot solve all the in-context retrieval tasks.
> 4. The proposed augmentations are closely related to the current empirical studies and can shed light on explaining empirical observations.
>    1. Theorem 5.2 (explicit retrieval augmentation) is an idealized version of a Retrieval-Augmented-Generation (RAG) system. Our result justifies theoretically that this method can enhance the representation power when the backbone model is an RNN.
>    2. Theorem 5.4 (hybrid model with single attention layer)  shows that few attention layers can greatly enhance the retrieval capability of RNNs. This is consistent with the empirical studies showing that merely 3 layers of attention can significantly improve the in-context retrieval of a 56-layer hybrid model \[1\].
>
> \[1\] An Empirical Study of Mamba-based Language Models
>
> **W4: Lack of discussion & comparison with RNN with ACT (Adaptive Computation Time)**
>
> We would like to thank the reviewer for pointing out the related work and we will discuss ACT in the related work section. However, a key difference between ACT and CoT is that ACT will not output intermediate thought tokens. This means that CoT may be a much more powerful decoding strategy that can significantly enhance the representation power. It has been shown by many previous theoretical works that CoT can enhance the representation power of Transformers \[1,2\], but we are not aware of similar analyses for ACT. The effectiveness of CoT on Transformers motivated us to study its effectiveness on RNNs, which had not yet been thoroughly studied before our work. Therefore, we disagree that our novelty is reduced by previous studies on ACT.
>
> \[1\] Towards Revealing the Mystery behind Chain of Thought: A Theoretical Perspective
> \[2\] Chain of Thought Empowers Transformers to Solve Inherently Serial Problems

---

> ### Author Response · Authors · 2024-11-26
> **Looking Forward to the Reviewer' s Reply**
>
> Dear reviewer,
>
> As the deadline approaches, we kindly ask if our responses have adequately addressed your concerns. We would greatly appreciate your feedback to ensure we have fully resolved any outstanding issues. In our response, we have clarified our contribution and we have especially stressed that we are comparing different architectures with the same number of **parameters**. We have also justified in detail why our results are not trivial.
>
> Thank you for your time and consideration.

---

> > ### Comment · Reviewer_bMYq · 2024-11-30
> >
> > Thank you for the detailed response and clarifications.
> >
> > > To overcome this limitation, the state size of RNN needs to scale linearly with respect to the context length for in-context retrieval.
> >
> > Totally agreed. However, the authors do not show that this holds for a standard RNN with O(N) state size and instead use this as an argument to claim that Transformers are more capable than RNNs. Instead, the authors propose augmenting RNNs with retrieval/attention, effectively increasing the total state size of the RNN+augmentation method to O(N).
> >
> > > The reason that we compare RNNs and Transformers with different state sizes is that RNNs indeed have smaller state sizes than Transformers with the same number of parameters.
> > > Our results are not tautological because not all approaches with O(n) memory can perform in-context retrieval.
> >
> > I agree that not all approaches with O(n) memory can perform in-context retrieval. The authors demonstrate two setups where this is achievable with O(n) memory. However, it is possible to consider a setup where both RNNs and Transformers have the same state size O(N) and the same number of parameters for a fixed input length N. In such a case:
> > - Would the RNN be incapable of performing in-context retrieval for lengths up to O(N)?
> > - Conversely, would the Transformer be able to perform in-context retrieval for lengths exceeding O(N)?
> >
> > If the answer suggests that an RNN with O(N) state size can solve in-context retrieval tasks as effectively as a Transformer, then the claim that RNNs are less capable than Transformers would not hold under equal state size constraints.
> >
> > > Our work reveals a fundamental limitation of this approach, showing that such models with high memory efficiency are incapable of doing in-context retrieval.
> >
> > The empirical limitations of RNNs in in-context retrival have already been demonstrated in prior works, including the references cited in the response and the paper. So, this one does not "reveal" this limitation.
> >
> > > We compare RNNs and Transformers with the same order of number of parameters because it is typical in current empirical studies.
> >
> > I agree on the importance of comparing models with the same order of parameters. However, for the in-context retrieval problem, state size is a crucial factor. If a model cannot fit the entire context into its state (or augmented state), how would it perform retrieval?
> >
> > >Models with larger state sizes do not automatically imply a stronger representation power.
> >
> > I agree with this statement, and I have no issue with the results and proofs that RNNs with O(log⁡n) state size are less capable than Transformers with O(n) state size. The issue, however, lies in the differing assumptions on state size. The key question remains unresolved: Can you demonstrate that RNNs and Transformers with the same state size would perform differently on in-context retrieval tasks?

---

> > > ### Author Response · Authors · 2024-12-01
> > >
> > > Thank you for your response. We would like to first clarify your question regarding the comparison criteria, i.e., whether we should compare different architectures with the same state size or the same parameter number.
> > >
> > > **Q1:** It is possible to consider a setup where both RNNs and Transformers have the same state size O(N) and the same number of parameters for a fixed input length N. In such a case:
> > >
> > > * Would the RNN be incapable of performing in-context retrieval for lengths up to O(N)?
> > > * Conversely, would the Transformer be able to perform in-context retrieval for lengths exceeding O(N)?
> > >
> > > Can you demonstrate that RNNs and Transformers with the same state size would perform differently on in-context retrieval tasks?
> > >
> > > **A1:**
> > >
> > > First, the recent trend of modern RNNs is based on the belief that RNNs can **save** memory for long inputs; therefore, **comparing the performance for the same state size is not a well-motivated setting**. In this case, RNNs are not more memory-efficient than Transformers and thus will not be preferred. A simple calculation can show that in this case, the inference cost for RNNs and Transformers will be in the same order as well. That is why we study the setting where both models have constant numbers of parameters, and under this setting, RNNs can save memory compared with Transformers for long inputs.
> > >
> > > Second, assuming that both RNNs and Transformers have numbers of parameters constant with input length, it is not possible to consider a setup where they also have the same state size.  This is because state size **is bounded by** the number of parameters for standard RNN architectures, including Mamba, LSTM, and Linear Transformer. In the meantime, the Transformer’s state size can be larger than the number of parameters when the context length grows. Since we are considering architectures with constant numbers of parameters and $O(\\log N)$-bit precision, RNNs and Transformers cannot have the same state size.
> > >
> > > Third, if we ignore the motivation and extend our hypothesis class to consider the setting where given a context length $N$, both RNNs and Transformers have the same state size O(N) and the same number of parameters, then both architectures can indeed perform in-context retrieval for context length $N$. However, the Transformer is still more capable because the same Transformer can perform in-context retrieval for a much longer context length (e.g. $\\Theta(N^3)$), but RNNs will fail.
> > > More specifically, we illustrate this point using the following simple example comparing Mamba and Transformers (the same argument also holds for RWKV and other architectures). We will show two points in response to the problems.
> > >
> > > 1. Given a context length $N$, if we consider a Mamba with the same state size and parameter as Transformers, both architectures can perform in-context retrieval for context length-$N$.
> > > 2. However, if we consider testing the two architectures above on a longer sequence with context length $N^3$, the Transformer architecture can still perform in-context retrieval but the Mamba will **fail** to do so.
> > >
> > > Given a context length $N$ and dimension $d$, the following tables show their corresponding parameters and state size per layer for both Mamba and Transformer
> > >
> > > | Architecture | \# Parameters per layer | State size per layer |
> > > | :---- | :---- | :---- |
> > > | Mamba | $\\Theta(d^2)$ | $\\Theta(d^2)$ |
> > > | Transformer | $\\Theta(d^2)$ | $\\Theta(dN)$ |
> > >
> > > 1. To align the parameters and state size for both architectures, we will need to choose the dimension for both architectures to be in the same order and the context length $N \= \\Theta(d)$.
> > >    1. This results in $N^2$ state sizes for both architectures.
> > >    2. Both architectures would be able to perform in-context retrieval on a context length $N$ given the state size.
> > > 2. However, if we consider the same architecture and test them on a longer sequence with sequence length $d^3$, using the same argument in our paper, we can show that Mamba will fail to perform in-context retrieval on this longer sequence but Transformers can still perform in-context retrieval. Actually, Transformers can even perform in-context retrieval for context length $\Theta(c^d)$ for some absolute constant $c \> 1$.
> > >
> > > Therefore, we conclude that if we fix the state size (although it is not well-motivated) in addition to fixing the parameter size and context length, then RNNs can match the performance of Transformers on the specific context length. However, once we stress-test the architectures with longer sequences, we can once again see the separation between RNNs and Transformers, as Transformers can dynamically enlarge their state size.

---

> > > > ### Author Response · Authors · 2024-12-01
> > > >
> > > > **Q2:** The empirical limitations of RNNs in in-context retrieval have already been demonstrated in prior works, including the references cited in the response and the paper. So, this one does not "reveal" this limitation.
> > > >
> > > > **A2:** We did not claim that our work is the first one to state that RNN has limitations in in-context retrieval. Our contribution is that our theory rigorously shows:
> > > >
> > > > 1. This limitation is fundamentally due to the limited state size of RNNs and is not amenable to methods like Chain-of-thought.
> > > > 2. Adding simple augmentations like in-context retrieval augmentation can bridge this representation gap. Notably, a minimal architecture change can work: adding a single Transformer layer to RNNs can bridge the gap. Note that this is more memory efficient than using a full Transformer that may have 100 layers.

---

> > > ### Author Response · Authors · 2024-12-02
> > >
> > > Dear Reviewer bMYq,
> > >
> > > As the discussion period deadline is today, we would like to know whether our response has adequately addressed your questions. We have laid out our arguments on why comparing models with the same state size, parameters, and context length is not well-motivated. We also provide a concrete example demonstrating that even under such an assumption, we will still observe the in-context retrieval gap between the two architectures once we test on longer sequences because the Transformer's state size dynamically increases with context length.
> > >
> > > Thank you for your time and consideration.

---

### Author Response · Authors · 2024-12-04

Dear AC,

We are reaching out to respond to the concerns raised by Reviewer bMYq. After an initial round of communication with the reviewer, we did not receive any further reply from the reviewer.

The reviewer suggested that we should compare models that have **the same number of parameters and the same state size for a specific context length**. In our responses, we provided comprehensive arguments explaining why such a comparison is not appropriate for our study:
1. The recent trend of modern RNNs is based on the belief that RNNs can **save memory** for long inputs; therefore, we are motivated to study the fundamental limitations of RNNs in the setting when they are more **memory-efficient** than Transformers. However, the reviewer is essentially proposing to study a setting where practitioners could not save memory by using RNNs.
2. We highlighted that comparing models under these constraints **does not account for the dynamic nature of the Transformer's state size**, since it can increase with the context length. We also included a concrete example demonstrating that even if we ignore the issue in motivation and compare RNNs and Transformers under the reviewer's proposed conditions, the in-context retrieval gap remains evident when tested on longer sequences.

We believe that the alternative setting proposed by the reviewer is not well-motivated and their concerns have been thoroughly addressed in our responses. We would like to draw the above discussion to your attention and sincerely hope you can take this into account when evaluating our paper.

Thank you for your time and consideration.

---

### Meta-Review · Area_Chair_zngA · 2024-12-19

**Metareview:**

This paper explores the representation capacity of memory efficient sequence models, and the effect resulting from the use of chain-of-thought techniques. Namely, authors show that RNNs cannot match the performance of Transformers even under chain-of-thought prompting. However, they do show that chain-of-thought strictly increases representation power. It's also shown that addressing the lack of ability of RNNs to perform in-context retrieval can close the gap with respect to transformers under chain-of-thought, which is done via hybrid architectures that include a transformer layer in a recurrent model.

The reviewers do not raise correctness concerns nor do they point out limitations with the tooling used for the analysis, so we assume conclusions hold. Instead, the discussion centered around fairness of comparisons in terms of equivalence of the architectures in parameter and state sizes, and on the relevance of the conclusions. To the first point, authors rebutted by arguing that the two architectures are fundamentally distinct and cannot be made completely equivalent, so they chose to match parameter sizes, as is commonly done in concurrent work. As per the relevance of the results, that boils down to opinion. I'd argue that highlighting fundamental limitations in RNNs is of great relevance to our community. There's been a recent surge in papers seeking to address known limitations in RNNs that would enable them reaching the level of performance we observe for quadratic self-attention models, and this work shows that this can only be achieved for long contexts if we somehow address the lack of ability of RNNs to perform in-context retrieval. While I do agree with some of the reviewers that claimed that the solution of adding linearly growing states to RNNs defeats the purpose of using an RNN in the first place, I see significant value in the negative results contributed by this work, which will help inform future work.

**Additional Comments On Reviewer Discussion:**

Reviewers mostly raised concerns regarding the fairness of comparisons and relevance of the results, to which authors responded satisfactorily in my opinion. Please see above for further detail.

---

### Decision · Program_Chairs · 2025-01-22

Accept (Poster)